# Adapters for Altering LLM Vocabularies: What Languages Benefit the Most?

**HyoJung Han**[‡]
University of Maryland
hjhan@cs.umd.edu

**Akiko I. Eriguchi**
Microsoft
akikoe@microsoft.com

**Haoran Xu**
Microsoft
haoranxu@microsoft.com

**Hieu Hoang**
Microsoft
hihoan@microsoft.com

**Marine Carpuat**
University of Maryland
marine@cs.umd.edu

**Huda Khayrallah**[‡]
Amazon
hudakh@amazon.com

## Abstract

Vocabulary adaptation, which integrates new vocabulary into pre-trained language models, enables expansion to new languages and mitigates token over-fragmentation. However, existing approaches are limited by their reliance on heuristics or external embeddings. We propose VocADT, a novel method for vocabulary adaptation using adapter modules that are trained to learn the optimal linear combination of existing embeddings while keeping the model's weights fixed. VocADT offers a flexible and scalable solution without depending on external resources or language constraints. Across 11 languages—with diverse scripts, resource availability, and fragmentation—we demonstrate that VocADT outperforms the original Mistral model (Jiang et al., 2023) and other baselines across various multilingual tasks including natural language understanding and machine translation. We find that Latin-script languages and highly fragmented languages benefit the most from vocabulary adaptation. We further fine-tune the adapted model on the generative task of machine translation and find that vocabulary adaptation is still beneficial after fine-tuning and that VocADT is the most effective.[1]

## 1 Introduction

Vocabulary adaptation (or transfer)—a process of modifying a pre-trained language model (LM) to use a new vocabulary—offers several key advantages. First, it enables the introduction of new languages into a model, increasing flexibility in handling linguistic diversity and improving downstream performance in target languages (Wang et al., 2020; Gogoulou et al., 2022; Downey et al., 2023). Second, it reduces over-fragmentation, where words are excessively split by the tokenizer, slowing down generation[2] and impairing performance in certain languages (Ahia et al., 2023; Petrov et al., 2023; Yamaguchi et al., 2024). These benefits have led to the development of numerous vocabulary adaptation approaches that initialize the new embeddings of new vocabulary with various methods based on heuristics (Mosin et al., 2023; Gee et al., 2022; Downey et al., 2023), external resources (Tran, 2020; Dobler & de Melo, 2023; Liu et al., 2024), or a separate hypernetwork that generates it (Minixhofer et al., 2024). They generally generate new embeddings using original embeddings, optionally followed by continued training to finalize the adaptation (Minixhofer et al., 2022; Ostendorff & Rehm, 2023; Dobler & de Melo, 2023; Liu et al., 2024).

However, existing vocabulary adaptation approaches face several limitations. Those that rely on heuristics (Gee et al., 2022; Downey et al., 2023), which use predefined rules to initialize new embeddings from existing ones rather than learning from data, often lack adaptability, where the new embeddings are not fully integrated into the original model and require an additional training phase of full-weight updates to fully adapt to the new vocabulary. Also, those that depend on external embeddings or networks (Tran, 2020; Dobler & de Melo, 2023; Liu et al., 2024), increase complexity

---

[‡]Work done at Microsoft.

[1]Project page: https://github.com/h-j-han/VocADT. Models at Huggingface Hub

[2]Standard transformer decoding is quadratic in sequence length, so length increases can be catastrophic.

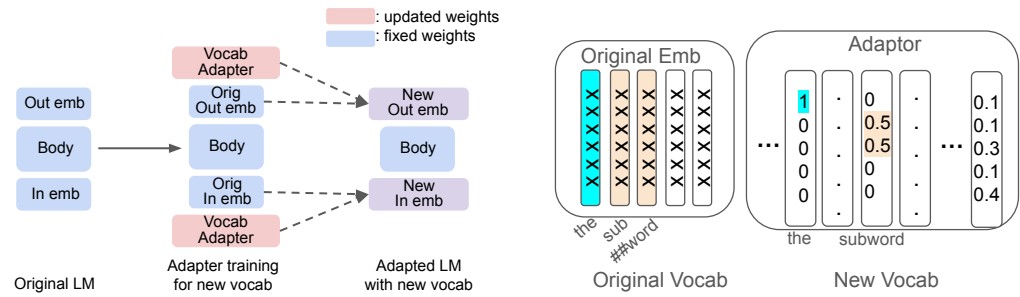

(a) Overview of the vocabulary adaptation and training.

(b) Initialization of vocabulary adapter.

Figure 1: Overview of our vocabulary adaptation with adapter (VocADT) and the initialization of adapter. The vocabulary adapter modules are trained to adapt new vocabulary with existing embeddings while keeping the original model fixed. We initialize entries of the adapter for overlapping tokens and tokens whose partitions are in the original vocabulary. Once trained, the adapters and original embeddings are merged to form the new embeddings.

and limit scalability. Furthermore, some approaches focus solely on language-specific cases or may have restrictions on the number of languages in the implementation when configuring the vocabulary (Dobler & de Melo, 2023; Minixhofer et al., 2024).

Additionally, we still know little about the impact of vocabulary adaptation across diverse linguistic and task settings. Most prior work investigates few languages, which is insufficient for identifying patterns (Tran, 2020; Ostendorff & Rehm, 2023; Remy et al., 2023; Yamaguchi et al., 2024), while studies that consider a broader range of languages only report averages without detailed analysis (Liu et al., 2024; Mundra et al., 2024). Furthermore, the impact of vocabulary adaptation on cross-lingual and generative tasks like machine translation (MT) is understudied, even though they represent crucial application areas for porting models to new languages. Many adaptation methods (Chung et al., 2020; Gee et al., 2022; Liu et al., 2024) have been evaluated instead on non-cross-lingual and discriminative tasks such as commonsense reasoning, natural language inference (NLI), or question answering (QA)—which are typically classification tasks.

We propose VocADT, a novel solution for vocabulary adaptation using adapters, designed to address key challenges in existing approaches (Figure 1). We introduce a vocabulary adapter module, a learnable matrix between the new vocabulary and the original embeddings of a language model. The module gradually adapts to new vocabularies through training while keeping all weights of the original model fixed, allowing the module to learn the best combination of the original embeddings without relying on heuristics, external embeddings, or dictionaries. This learned adaptation approach offers better adaptability of new embeddings to the original language model with only its embeddings replaced and more flexibility in the number of languages while removing the necessity of external pre-trained resources. At the end of training, the adapter is merged with the original embeddings to create a new embedding matrix.

In addition to our novel method, we empirically address the following key questions to understand the effectiveness and behavior of vocabulary adaptation: (1) Which languages benefit most from vocabulary adaptation? (2) What are the best strategies for creating new vocabularies? Also, is script consistency necessary? (3) How does vocabulary adaptation impact machine translation? We emphasize this task as it is a critical task for multilingual models that involves cross-lingual and generative capabilities, which are often more complex than classification or monolingual tasks.

We demonstrate the effectiveness of our adaptation method on various NLP tasks spanning Natural Language Understanding and MT. Results show that our approach consistently surpasses the original Mistral model in most cases, both after the adaptation phase and following phase of full-weight training. Additionally, our method outperforms or matches other strong vocabulary adaptation baselines. Our findings indicate that Latin-script languages and those with severe fragmentation benefit the most from vocabulary adaptation. Finally, while all vocabulary adaptation methods continue to be effective for machine translation after fine-tuning, VocADT shows the best results among them. Our main contributions are summarized as follows:

- We propose VocADT, a simple and effective solution for vocabulary adaptation using adapters, that addresses key limitations in prior work such as reliance on external embedding or language constraints.

- We conduct experiments that cover a wide range of languages and scripts, finding that languages with Latin scripts or severe fragmentation benefit the most and that having a consistent grouping of scripts for multilingual vocabulary is helpful.

- Our approach consistently outperforms the original language model and is more effective than, or on par with, strong vocabulary adaptation baselines after the adaptation phase across various tasks and after the following full-weight fine-tuning on MT.

## 2 BACKGROUND AND MOTIVATION

**Approaches to Vocabulary Adaptation**   Prior work focuses on initialization strategies for the new vocabulary embeddings, before continuing training with unlabeled target language text using the original self-supervised pretraining objective. For instance, FOCUS (Dobler & de Melo, 2023) initializes embeddings as a weighted combination of overlapping tokens using external embeddings for non-overlapping ones, while copying the embeddings of overlapping tokens. OFA (Liu et al., 2024) also relies on external word vectors to initialize embeddings for non-shared new tokens, using a weighted average of original tokens based on semantic similarity. This strategy often requires external resources such as auxiliary embeddings (Dobler & de Melo, 2023; Liu et al., 2024; Ostendorff & Rehm, 2023) or bilingual dictionaries (Mundra et al., 2024; Minixhofer et al., 2022).

After initialization, language adaptive pretraining (LAPT; Chau et al., 2020) usually updates all model weights (Tran, 2020; Liu et al., 2024; Dobler & de Melo, 2023; Ostendorff & Rehm, 2023), except Yamaguchi et al. (2024) which use LoRA (Hu et al., 2022). Downey et al. (2023) show full-weight updates outperform embedding-only training, which is insufficient for multilingual transfer.

Other vocabulary adaptation strategies introduce architecture-specific changes to the model, such as MAD-X (Pfeiffer et al., 2020), which incorporates various adapters into Transformer models, and thus additional computation costs. There are few alternatives to these resource-intensive approaches. A notable exception is ZeTT (Minixhofer et al., 2024), which trains a hypernetwork that generates embeddings for the new vocabulary, allowing immediate zero-shot use by only replacing embeddings without further model training. It can be extended to multilingual hypernetworks by appending a learnable language-specific embedding.

**Linear Combination of Embeddings**   Most vocabulary transfer methods combine the existing embeddings to generate new ones. A popular approach is to use a weighted average of the original embeddings (**bolded** in Appendix Table 9). For example, Gee et al. (2022) and Mosin et al. (2023) compute the new embeddings by simply averaging the embeddings of subword tokens, while Tran (2020), Minixhofer et al. (2022), OFA, and FOCUS utilize external resources to determine the weights to initialize the new embeddings with a weighted average of the original embeddings. Mundra et al. (2024) established theoretically that initializing within the convex hull of existing embeddings—e.g., using a weighted average of source embeddings—is a good initialization.

Our motivation stems from the question: rather than deciding how to combine existing embedding vectors heuristically, why not learn this process to create new embedding vectors? Relying on heuristics may lack adaptability that typically requires an additional training phase of full-weight updates to fully adapt to the new vocabulary. Building upon prior works, we propose to learn linear combinations with vocabulary adapters.

**Empirical Evaluations**   Many language adaptation experiments have been conducted using new language-specific monolingual vocabularies (Minixhofer et al., 2024; Dobler & de Melo, 2023; Pfeiffer et al., 2020; Minixhofer et al., 2022; Yamaguchi et al., 2024), as well as English-only but domain-specific vocabularies (Gee et al., 2022; Mosin et al., 2023). In contrast, Liu et al. (2024) and Mundra et al. (2024) use a single unified multilingual vocabulary covering at least 369 languages and four languages, respectively.

Downey et al. (2023) conducted experiments with both monolingual and multilingual vocabularies across eight languages and additional vocabulary from the Uralic family. While their findings

indicated that multilingual adaptation in the Uralic family followed overall trends, it remains unclear whether vocabulary adaptation benefits languages in different script groups. Overall, empirical evidence is still lacking to guide practical decisions for grouping languages in multilingual models.

Furthermore, most studies exclusively evaluate models on non-cross-lingual and non-generative tasks, such as binary or multi-class classification, sequence labeling (e.g., part-of-speech tagging), or answer span prediction. Mundra et al. (2024) and Yamaguchi et al. (2024) include monolingual generative tasks like summarization. As a result, the impact of vocabulary adaption on cross-lingual generation tasks such as MT remains understudied, even though this is a crucial application area. [3]

To address these gaps, this work introduces a strategy to adapt LLM to new vocabularies, and experiments designed to measure its impact across diverse linguistic and task settings. We compare empirically against FOCUS and OFA, representing more resource-intensive initialization approaches, and ZeTT, representing a more parsimonious approach that has only been tested on limited tasks so far. Summaries of prior work can be found in Appendix Table 9.

# 3  VOCADT: MULTILINGUAL VOCABULARY ADAPTATION WITH ADAPTERS

In this section, we outline our approach to multilingual vocabulary adaptation using adapter modules (VocADT). We detail the architecture of the adapter module (§3.1) and the initialization process (§3.2). Additionally, we introduce an additional loss for handling overlapping tokens between the new and original vocabularies (§3.3). Finally, we further fine-tune the vocabulary adapted model for downstream task (§3.4).

## 3.1  VOCABULARY ADAPTER MODULE

We introduce the vocabulary adapter modules to find parameters of new embeddings that can replace the original embedding without changing the non-embedding part of the original model (Figure 1a). For simplicity, we refer to both input and output embeddings (or the LM head) collectively as embeddings. Let $V^o$ and $V^n$ be the *o*riginal and *n*ew vocabulary, respectively, and let $\mathcal{T}^x : w \to (t_1, t_2, \ldots, t_k)$ be a tokenizer associated with a vocabulary $V^x$ where $t_j \in V^x, \forall j = 1, \ldots, k$. We put vocabulary adapter modules $\boldsymbol{A} \in \mathbb{R}^{|V^n| \times |V^o|}$ between the new vocabulary $V^n$ and the original embedding $\boldsymbol{E}^o \in \mathbb{R}^{|V^o| \times h}$ where $h$ is an embedding dimension, in a manner similar to bottleneck adapters (Houlsby et al., 2019). We train the adapters with the standard language modeling loss $\mathcal{L}^{lm}$, where we freeze the original weights and only update the adapters. This may be analogous to finding the new embedding vector for a token with the weighted combination of original embedding vectors (Downey et al., 2023; Dobler & de Melo, 2023; Liu et al., 2024). Unlike similar works, our approach learns the weights for embedding combination. After training the adapters, we get new embeddings $\boldsymbol{E}^n \in \mathbb{R}^{|V^n| \times h}$ by merging the original embeddings and adapters to $\boldsymbol{E}^n = \boldsymbol{A}\boldsymbol{E}^o$, which results in a language model with the same architecture as the original one but with a different vocabulary size.

## 3.2  INITIALIZING ADAPTER

Effective initialization of the new embedding is crucial in adapting to a new vocabulary, as fully random initialization is widely recognized for leading to poor performance (Minixhofer et al., 2022; Yamaguchi et al., 2024). In our case, random initialization of the adapter $\boldsymbol{A}^0$ is equivalent to random initialization of $\boldsymbol{E}^n$, making proper initialization of $\boldsymbol{A}^0$ equally important. We suggest a simple initialization scheme for the vocabulary adapter, illustrated in Figure 1b.

First, we follow the common methods of copying the original embeddings of overlapping tokens by setting a one-hot vector in the adapter. Let $\mathcal{I}^x : V^x \to \mathbb{Z}$ be the mapping function of a token to an index in a vocabulary $V^x$ and let $i = \mathcal{I}^n(w)$ be the index of a token $w$ in $V^n$. The row of the adapter $\boldsymbol{A}^0_i$ corresponding to the overlapping tokens $w \in V^o \cap V^n$ is set as follows:

$$\boldsymbol{A}^0_{i,\mathcal{I}^o(w)} = 1, \quad \boldsymbol{A}_{i,j} = 0 \quad \forall j \neq \mathcal{I}^o(w), \quad \text{where } w \in V^o \cap V^n. \tag{1}$$

---

[3]Mundra et al. (2024) report MT results, but they do not release the models or per-language performance metrics, making direct comparisons difficult.

Inspired by Gee et al. (2022), we then initialize the row of a token $w$ in $\boldsymbol{A}^0$, whose partitioned tokens by the original tokenizer $\mathcal{T}^o$ are subset of the original vocabulary, $\mathcal{T}^o(w) = \{t_1, \ldots, t_m\} \subset V^o, m > 1$, with normalized multi-hot vector as below. This corresponds to directly initializing new embedding with the average of the original embeddings associated with the tokens produced by $\mathcal{T}^o$.

$$\boldsymbol{A}^0_{i,j} = \begin{cases} \frac{1}{m} & \text{if } j \in \{\mathcal{I}^o(t_1), \ldots, \mathcal{I}^o(t_m)\} \\ 0 & \text{otherwise} \end{cases} \quad \text{where } \begin{aligned} & w \in V^n \backslash (V^o \cap V^n) \quad \text{and} \\ & w \in S = \{w \mid \mathcal{T}^o(w) = \{t_{1:m}\} \subset V^o\}. \end{aligned} \quad (2)$$

For a token that does not fall into the first two cases above (i.e. a non-overlapping token and its partitions by $\mathcal{T}^o$ are not in $V^o$), we randomly initialize a row vector of the adapter with the uniform distribution whose sum of each element is one as follows:

$$\boldsymbol{A}^0_i = \frac{\mathbf{u}}{\sum_{j=1}^{|V^o|} u_j}, \quad u_j \sim \text{Uniform}(0,1), \; j = 1, \ldots, |V^o| \quad \text{where } w \in V^n \backslash (V^o \cap V^n) \backslash S. \quad (3)$$

### 3.3 AUXILIARY LOSS

As training progresses, the adapter entries of overlapped tokens tend to diverge from their initial states. This divergence can be undesirable because the original embeddings are already well-integrated into the language model, and our goal is more focused on adjusting the embeddings of the newly introduced vocabulary items. Following Minixhofer et al. (2024), we experiment with an additional loss term that encourages the adapter entries for overlapping words to remain close to their initial values, formulated as follows:

$$\mathcal{L}^{aux} = \frac{1}{|V^o \cap V^n|} \sum_{w \in |V^o \cap V^n|} ||\boldsymbol{A}_{\mathcal{I}^n(w)} - \boldsymbol{A}^0_{\mathcal{I}^n(w)}||_2. \quad (4)$$

The final loss for the adapter training is the combination of the standard language loss and additional loss with the weighing factor of $\alpha$, $\mathcal{L}^{tot} = \mathcal{L}^{lm} + \alpha \mathcal{L}^{aux}$.

### 3.4 FURTHER FINE-TUNING FOR DOWNSTREAM TASK

To understand the impact of vocabulary adaptation after task-specific fine-tuning, we follow the full ALMA (Xu et al., 2024) training on all model parameters for the cross-lingual generation task of machine translation after our VocADT on just the embeddings. ALMA training begins with fine-tuning on monolingual data, followed by further weight optimization on small curated parallel data.

## 4 WHICH LANGUAGES BENEFIT THE MOST FROM VOCABULARY ADAPTATION?

We aim to understand "When and how should we perform vocabulary adaptation?". More specifically, we seek insight into which languages might benefit the most from vocabulary adaptation in terms of improving overall performance or mitigating over-fragmentation.[4]

To this end, we design experiments to cover 10 non-English languages along with English, listed in Table 1, with a variety of scripts and language families. These languages are broadly categorized into three groups: (1) *Latin* group of Swahili, Indonesian, Estonian, and Haitian, which are low- to mid-resource languages and all use Latin script; (2) *Mixed* group including Korean, Greek, Russian, and Bulgarian, which utilize a mixture of scripts; (3) *Cyrillic* group for languages with that scrip.[5]

We test individual language adaptation with language-specific vocabularies. We also adapt several multilingual vocabularies that include English and four non-English languages in a single shared vocabulary, with each group corresponding to one of the previously mentioned groups. This is to identify a language grouping strategy—whether to mix languages with different scripts or grouping languages with consistent scripts.

---

[4]The non-English languages that we cover are all highly fragmented by common LLMs, and their fragmentation is similarly improved by our method. Therefore, our analysis focuses on performance.

[5]In Section 6.4 and Appendix F.1, we experiment with *All* group including all languages mentioned here.

Table 1: Covered Languages and its availability in multilingual benchmarks. We mainly categorize non-English languages by scripts—*Latin* group (2-5) and the *Mixed* group (6-7). We additionally experiment with *Cyrillic* group (8-11). We follow the resource-level of languages from Joshi et al. (2020) and Üstün et al. (2024)

| idx | Full Name | Short | Script | Resource | FLORES | XNLI | XCOPA | Belebele | MMMLU |
|-----|-----------|-------|--------|----------|--------|------|-------|----------|-------|
| 1 | English | en | Latin | High | ✓ | ✓ | | ✓ | ✓ |
| 2 | Swahili | sw | Latin | Low | ✓ | ✓ | ✓ | ✓ | |
| 3 | Indonesian | id | Latin | Mid | ✓ | | ✓ | ✓ | ✓ |
| 4 | Estonian | et | Latin | Mid | ✓ | | ✓ | ✓ | |
| 5 | Haitian Creole | ht | Latin | Low | ✓ | | ✓ | ✓ | |
| 6 | Korean | ko | Hangul | High | ✓ | | | ✓ | |
| 7 | Greek | el | Greek | Mid | ✓ | ✓ | | ✓ | |
| 8 | Russian | ru | Cyrillic | High | ✓ | ✓ | | ✓ | ✓ |
| 9 | Bulgarian | bg | Cyrillic | Mid | ✓ | ✓ | | ✓ | |
| 10 | Ukrainian | uk | Cyrillic | Mid | ✓ | | | ✓ | ✓ |
| 11 | Kazakh | kk | Cyrillic | Mid | ✓ | | | ✓ | |

## 5 EXPERIMENT DESIGN

### 5.1 BASELINES AND MODELING

We use Mistral-7B (Jiang et al., 2023) as our language model, along with its original vocabulary, which consists of 32k tokens ($|V^o| = 32k$). As baselines, we evaluate three state-of-the-art methods for vocabulary adaptation, ZeTT (Minixhofer et al., 2024), FOCUS (Dobler & de Melo, 2023), and OFA (Liu et al., 2024). For ZeTT and FOCUS, we experiment with language-specific vocabularies (ZeTT-mono, FOCUS-mono) as their implementations require specifying the language to adapt the vocabulary. This results in separate adaptations per language, which could be hard to scale with larger language coverage.[6] For VocADT and OFA methods (VocADT-multi, OFA-multi), we experiment with multilingual vocabularies of five languages including English and four non-English languages, where we define three distinct language groups such as {en, sw, id, et, ht} ( *Latin* group), {en, ko, el, ru, bg} (*Mixed* group), and {en, ru, bg, uk, kk} (*Cyrillic* group).

### 5.2 TRAINING VOCADT

**Vocabulary.** We train SentencePiece (Kudo & Richardson, 2018) tokenizers on either language-specific corpora or a combined corpus, with a maximum of 2 million tokens per language, and create new vocabularies with a size of 50k for all cases including mono/multilingual vocabularies ($|V^n| = 50k$). Newly created vocabularies for each language group are shared across baselines.

**Adapter Training.** In the adapter training phase, we train only the adapters, while fixing all parameters of the original model. The input and output adapters are separate modules, as preliminary results showed that sharing an adapter for the input and output sides performs worse. We train 0.5B monolingual tokens per language, totaling 2.5B mixed by 5 languages (English + 4 non-English from each corresponding group), and report test numbers from it. We use "clean" documents from the corpus of MADLAD-400 (Kudugunta et al., 2023). We set the weighing factor of auxiliary loss $\alpha$ with 0.1 for non-Latin groups and 0 for the *Latin* group unless otherwise specified. This is based on the empirical results in Appendix A that maintaining the embeddings of overlapping tokens close to the original status during the adaptation is effective only for non-Latin script languages and counter-effective for Latin languages. More details regarding the training are in Appendix C.

### 5.3 FULL-WEIGHT FINE-TUNING

After the adaptation phase, we follow the fine-tuning recipe of ALMA (Xu et al., 2024) that consists of full-weight training with monolingual corpus and a small amount of high-quality parallel corpus

---

[6]ZeTT does not support Ukrainian and Kazakh, therefore we primarily compare and average the results for 9 languages covered by both methods. See Appendix E for results for Ukrainian and Kazakh.

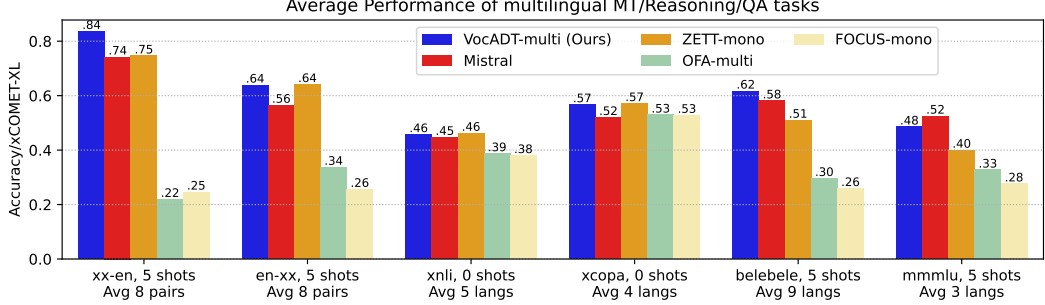

Figure 2: Average scores of original Mistral and its adaptation with new vocabulary, only replacing embeddings and fixing the body of transformer modules. "-multi" indicates models with a multilingual vocabulary, which includes five languages covering all languages with two separate models, while "-mono" refers to monolingual vocabulary models. `xx-en` and `en-xx` indicate MT tasks. See Appendix E for individual values.

to enhance MT performance (§3.4). We also include Mistral in this phase of training. 1) In monolingual fine-tuning, we use MADLAD-400. For adapted ZeTT and FOCUS models (prior work), we fine-tune each separate model with non-English language-specific vocabulary except for `uk` and `kk` for ZeTT (again, due to unsupported languages in ZeTT) using a total of 2B tokens combining English and the corresponding non-English. For Mistral and the adapted VocADT and OFA, we fine-tune separate models for all three non-English groups (*Latin*, *Mixed*, *Cyrillic*) plus English using a corpus of 5B monolingual tokens containing 5 languages. 2) In the next parallel training, we sample 15k bitext from NLLB dataset (Schwenk et al., 2021b; Heffernan et al., 2022; NLLB Team et al., 2022)[7] for each English and non-English training pairs with top LASER3 scores (Artetxe & Schwenk, 2019). The parallel training is done for one epoch, and we report test set numbers with the best model of the validation set. All the models are fine-tuned and tested with both directions of `en-xx` and `xx-en` within a single model, meaning there are no separate models for opposite translation directions. We follow the prompting strategy of Xu et al. (2024).

### 5.4 EVALUATION

We evaluate adaptation methods with multilingual benchmarks of various tasks including MT, natural language inference (NLI), common sense reasoning, and multiple choice question answering (QA). For MT of English to non-English (`en-xx`) and non-English to English (`xx-en`), we use FLORES (Goyal et al., 2022; NLLB Team et al., 2022) as it supports all the languages that we experiment with. We use five-shot MT prompting for the model from the adaptation phase, and zero-shot prompting for the model after the ALMA training phase. We assess the translation quality with xCOMET-XL (Guerreiro et al., 2023), which produces a score of increasing quality ranging from 0 to 1. For NLI and reasoning, we use XNLI (Conneau et al., 2018) and XCOPA (Ponti et al., 2020) with zero-shot prompting. For multiple choice QA, we use Belebele (Bandarkar et al., 2024) and Multilingual MMLU (Hendrycks et al., 2021; Lai et al., 2023, MMMLU) with five shot prompting. All the tasks except for MT are classification tasks, where we use the `lm-evaluation-harness` (Gao et al., 2024) evaluation tool and report accuracy.

## 6 VOCABULARY ADAPTATION RESULTS AND ANALYSES

### 6.1 OVERALL TASK PERFORMANCE

We first present the controlled comparison on diverse tasks of the original Mistral with new vocabulary variants obtained by our vocabulary adaptation approach (VocADT) and the ZeTT and OFA baselines. Figure 2 presents the average performance across multiple multilingual MT, NLI, reasoning, and QA tasks. Language-wise results are in Appendix E. Overall, adapting the vocabulary using

---

[7]https://huggingface.co/datasets/allenai/nllb

VocADT generally leads to better performance compared to the original Mistral model, and either surpasses or performs on par with ZeTT. MMMLU is the only task where Mistral still holds the top spot; however, the performance gap between the new and original embeddings is smaller with VocADT approach than with ZeTT. Remarkably, VocADT-multi achieves these results with only two models for the eight languages tested, whereas ZeTT requires a separate model for each language.

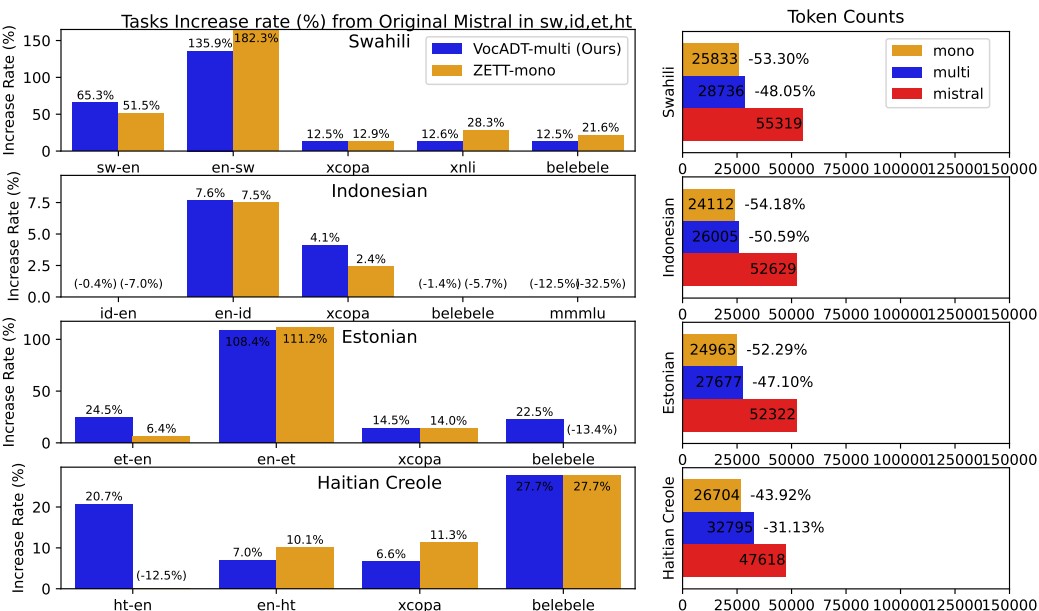

(a) Increase rate of task performance and number of token count of *Latin* group languages.

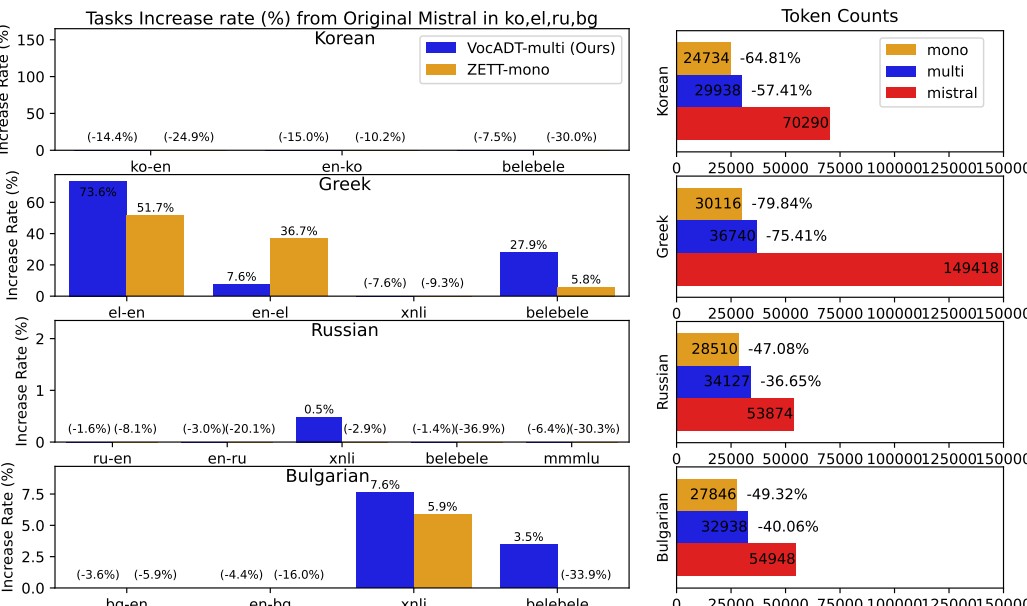

(b) Increase rate of task performance and number of token count of *Mixed* group languages.

Figure 3: Effect of vocabulary adaption on mitigating over-fragmentation and task performance. The $y$-axis for the increase rate on the left side is limited to the positive range. Languages with Latin scripts or those experiencing severe fragmentation benefit the most. `xx-en` and `en-xx` are machine translation tasks. See Appendix E for individual task performance values.

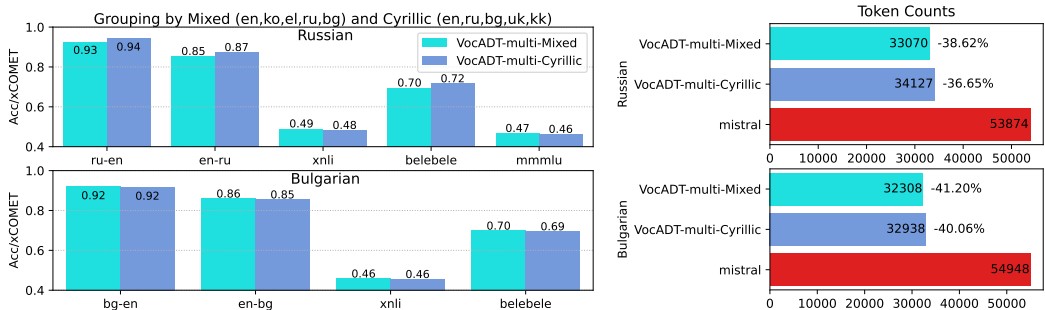

Figure 4: Comparison of task performance between two grouping strategies of *Mixed*-script and *Cyrillic*-script on two shared languages. Consistent script within a group provides minor benefits.

## 6.2   WHICH LANGUAGES BENEFIT THE MOST FROM VOCABULARY ADAPTATION?

The previous section presented a macro view of performance across all languages. This section drills down to look on the impact of vocabulary adaptation in a language-wise manner. Figure 3 shows the increased rate of task performance and tokenization statistics for various languages after applying different vocabulary adaptation methods. The results are shown for *Latin* group languages (Swahili, Indonesian, Estonian, and Haitian Creole) and *Mixed* group (Korean, Greek, Russian, and Bulgarian). We compare VocADT and ZeTT against the original Mistral model. The vertical axis of the increase rate is fixed to the positive range to see the benefit trends more easily and all the numbers including the negative range of the increase rate are in Appendix E. We use the FLORES development set for counting the tokens by various tokenizers where the semantic contents of every language are the same.

**Languages with Latin Scripts or Severe Fragmentation Benefit the Most**   In Figures 3a, we observe that Latin script languages consistently benefit from vocabulary adaptation, regardless of the task, adaptation method employed. However, even non-Latin languages show improvements when they suffer from severe over-fragmentation, as seen in the case of Greek in Figures 3b. Among the eight languages, Greek is the most fragmented by the Mistral tokenizer, and it demonstrates significant improvement after the adaptation to less fragmented vocabulary, particularly in MT tasks, while other non-Latin languages in *Mixed* group show zero, modest, or even negative gains.

In Appendix B, we further discuss the pronounced performance declines observed in Korean compared to Russian or Bulgarian within the same Mixed group. Despite improving fragmentation for Korean, we suspect that the linear combination assumption is insufficient given the lack of representation of Korean characters in the original vocabulary.

## 6.3   DOES SCRIPT MATTER FOR LANGUAGE GROUPING?

Multilingual vocabularies for language groups can strike a balance between the extensive coverage of the original Mistral and the limited scope of language-specific monolingual models. We investigate strategies for grouping, in particular the effect of script.

Figure 4 compares the performance and token count reduction between two non-English grouping strategies for Russian and Bulgarian: *Mixed*-script (`ko`, `el`, `ru`, `bg`) and *Cyrillic*-script (`ru`, `bg`, `uk`, `kk`) languages. For Russian, the consistent script language group performs slightly better, especially in the MT task. For Bulgarian, both grouping strategies deliver nearly identical results. Overall, the results suggest that maintaining a consistent script within a group enhances performance, though outcomes tend to be influenced more by the language type itself than by the grouping strategy.

## 6.4   SCALABILITY AND GENERALIZABILITY OF VOCADT

We further explore the language scalability of the method with *All* language groups including 11 languages (§F.1), and the generalizability of our VocADT findings to other language models (§F.2). Both scalability and generalizability experiments show that the *All* group follows trends similar

to the *Latin*, *Mixed*, and *Cyrillic* setups while the performance trends observed with LLaMA are consistent with those seen in Mistral.

# 7 IMPACT OF VOCABULARY ADAPTATION ON DOWNSTREAM FINE-TUNING

Do the effects of vocabulary adaptation hold up after fine-tuning the adapted language model? After completing the VocADT process, which keeps non-embedding model weights fixed, we update the full weights of the adapted model to enhance MT performance following ALMA (Xu et al., 2024).

As can be seen in Table 2, all vocabulary adaptation approaches are effective compared to Mistral except for `en-sw`, and among those, our approach (VocADT) achieves the highest average score in both `en-xx` and `xx-en` directions. In the `xx-en` direction, the performance of VocADT matches that of ZeTT, despite using a smaller number of individual models of the same size (2 VocADT vs 8 ZeTT). Interestingly, language-specific models (ZeTT, FOCUS) tend to excel in Latin languages, whereas multilingual models (Mistral, VocADT, OFA) generally outperform language-specific models in non-Latin cases.

In sum, with full parameter fine-tuning after the vocabulary adaptation, our VocADT model offers a competitive edge across both `xx-en` and `en-xx` tasks, further validating the effectiveness of our approach. VocADT demonstrates that a multilingual model can achieve or surpass language-specific models like ZeTT, offering a more flexible and scalable solution for handling multiple languages.

Table 2: MT performance after full-weight fine-tuning the new vocabulary-adapted model. The symbol "#" indicates the number of separate models for this experiment table. All vocabulary adaptation approaches after fine-tuning are effective compared to Mistral except for `en-sw`. VocADT-multi shows the best average performance in both directions while matching the score of ZeTT in `xx-en`.

| FLORES | xx-en | | | | | en-xx | | | | |
| Lang (*group*) ↓ | VocADT | Mistral | ZeTT | OFA | FOCUS | VocADT | Mistral | ZeTT | OFA | FOCUS |
|---|---|---|---|---|---|---|---|---|---|---|
| sw (*Latin*) | 0.893 | 0.891 | **0.897** | 0.889 | 0.893 | 0.753 | **0.770** | 0.762 | 0.763 | 0.762 |
| id (*Latin*) | 0.951 | 0.950 | **0.953** | 0.951 | 0.945 | 0.874 | 0.874 | **0.879** | 0.871 | 0.876 |
| et (*Latin*) | 0.939 | 0.925 | **0.941** | 0.937 | 0.939 | 0.852 | 0.868 | 0.856 | 0.845 | **0.870** |
| ht (*Latin*) | **0.706** | 0.696 | 0.699 | 0.699 | 0.685 | 0.336 | 0.336 | 0.334 | 0.332 | **0.339** |
| ko (*Mixed*) | 0.886 | 0.892 | 0.898 | 0.897 | **0.906** | 0.755 | 0.715 | **0.772** | 0.742 | 0.754 |
| el (*Mixed*) | **0.922** | 0.845 | 0.910 | 0.894 | 0.902 | **0.876** | 0.817 | 0.857 | 0.862 | 0.861 |
| ru (*Mixed*) | 0.945 | 0.894 | 0.944 | **0.946** | 0.945 | **0.889** | 0.828 | 0.872 | 0.882 | 0.868 |
| bg (*Mixed*) | **0.953** | 0.904 | 0.952 | 0.950 | 0.949 | 0.895 | 0.844 | 0.895 | **0.899** | 0.895 |
| Avg (8 pairs) | **0.899** | 0.875 | **0.899** | 0.895 | 0.895 | **0.779** | 0.757 | 0.778 | 0.774 | 0.778 |
| uk (*Cyrillic*) | **0.941** | **0.941** | _ | 0.936 | 0.928 | **0.878** | 0.876 | _ | 0.868 | 0.840 |
| kk (*Cyrillic*) | **0.881** | 0.875 | _ | 0.875 | 0.880 | **0.807** | 0.790 | _ | 0.785 | 0.796 |
| Avg (10 pairs) | **0.902** | 0.881 | _ | 0.897 | 0.897 | **0.792** | 0.772 | _ | 0.785 | 0.786 |
| # of Models → | 3 | 3 | 8 | 3 | 10 | 3 | 3 | 8 | 3 | 10 |

# 8 CONCLUSION

We propose a simple and effective vocabulary adaptation method using a vocabulary adapter. Our approach consistently outperforms the original Mistral model after the adaptation phase across various tasks and after the following full-weight finetuning on machine translation. Furthermore, our method is on par with or more effective than strong vocabulary adaptation baselines, without relying on external embeddings or language constraints, offering a flexible and scalable solution for handling multiple languages. Our experiments cover a wide range of languages and scripts, revealing that languages with Latin scripts or severe fragmentation benefit the most. We also explored different grouping strategies, finding that maintaining consistent scripts within a group offers relatively minor benefits. Lastly, with a focus on machine translation, we confirm that vocabulary adaptation remains effective even after full-weight fine-tuning, and VocADT is the most effective approach.

ACKNOWLEDGEMENTS

We thank Anthony Aue for early discussions, and Marcin Junczys-Dowmunt and the anonymous reviewers for their insightful and helpful feedback.

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

## A    IS THE AUXILIARY LOSS HELPFUL?

We examine the effects of the auxiliary loss that aims to mitigate the divergence of original status for overlapping words as described in Section 3.3. Figure 5 illustrates the impact on task performance of vocabulary adaptation with and without auxiliary loss on *Latin* and *Mixed* group vocabulary. We report the average of four non-English languages in each group along with English.

For *Latin* languages (left plot of Figure 5), omitting the auxiliary loss ($\alpha = 0$) performs slightly better or comparably to using a non-zero $\alpha$. For the *Mixed* group plus English vocabulary (right plot of Figure 5), maintaining the embedding values of overlapping words shows slight effectiveness in both non-English and English. We hypothesize that non-Latin languages are less prone to have word collisions with the original vocabulary compared to the *Latin* group, as the Mistral model is largely English (Latin) centric. As a result, retaining the established embeddings for overlapped words in *Latin* group vocabulary and Mistral vocabulary may disrupt effective learning due to the possible similarity in scripts with English. On the other hand, keeping the original embeddings during the adaptation for overlapping tokens in *Mixed* may be helpful to maintain the already established embeddings for overlapped tokens while adjusting the embeddings for new non-Latin script tokens.

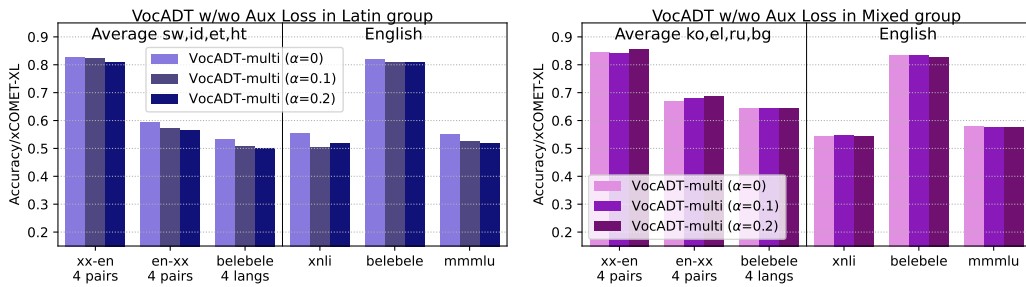

Figure 5: Effects of the auxiliary loss on various settings.

This auxiliary loss can be thought of as a form of regularization. There is a long history of applying regularization during adaptation of MT models to control the adaptation process by limiting the amount that the output distribution or the weights of the fine-tuned model can vary from the original model weights. Prior work has explored using dropout and L2 regularization (Miceli Barone et al., 2017), cross entropy (Khayrallah et al., 2018), freezing parts of the network (Wuebker et al., 2018; Thompson et al., 2018), and Elastic Weight Consolidation (Kirkpatrick et al., 2017; Thompson et al., 2019a;b). As our auxiliary loss has mixed effectiveness depending on language characteristics, future work could consider other methods.

## B    DISCUSSIONS ON NON-ALPHABETIC SCRIPTS AND POSSIBLE LIMITATIONS OF LINEAR COMBINATION ASSUMPTION

As shown in Figure 3b, the performance decline for Korean in VocADT vocabulary transfer (-14% to -15% in MT) is more pronounced than for Russian or Bulgarian (-1% to -4%) within the same Mixed group, even though the fragmentation improvement for Korean is greater. This deviates from the expected performance improvements seen in Greek, where mitigating extreme over-fragmentation (150k tokens) led to gains. Although Korean has significant over-fragmentation (70k tokens), its severity is less than half that of Greek and closer to Russian and Bulgarian (54k tokens).

One possible explanation is the limitation of our assumption that new embeddings can be solely represented as linear combinations of old embeddings. This assumption may not hold well for Korean, which uses the non-alphabetic Hangul script. In Hangul, tokens represent entire syllables or consonant-vowel combinations rather than individual phonemes, making them difficult to decompose into subwords using the original tokenizer. For instance, the new token "처럼" (meaning "like" in English) appears frequently in Korean. However, the original Mistral vocabulary lacks a dedicated token for "럼", preventing proper decomposition of "처럼" without resorting to byte-level tokens. These byte-level fallbacks may not effectively capture the linguistic structure of the character, potentially degrading performance. This issue is less prevalent in alphabetic scripts such as Latin, Cyrillic,

or Greek, where words can be easily broken down into individual characters. This limitation may account for the observed performance discrepancy.

## C   DETAILS OF TRAINING, BASELINE & EVALUATION

Here we describe training in detail. We use four Nvidia A100 GPUs for adapter training and 16 AMD MI200 GPUs for full-weight fine-tuning. For all monolingual training including adaptation phase and fine-tuning phase, we follow (Xu et al., 2024) for setting the sampling ratio of monolingual data to mitigate language balance in the monolingual data and avoid prioritizing English. The method fixes the sampling ratio for English with a certain probability e.g. $1/n$ if there are $n$ languages to mix and allocate the remaining ratio (e.g. $\frac{n-1}{n}$) by employing temperature sampling suggested by Aharoni et al. (2019). We mix the monolingual data for *Latin* group {en, sw, id, et, ht} with {17%, 16%, 32%, 23%, 12%} ratio, for *Mixed* group {en, ko, el, ru, bg} with {17%, 17%, 19%, 30%, 17%}, and for *Cyrillic* {en, ru, bg, uk, kk} with {17%, 32%, 18%, 20%, 13%}. For *Add* group in §6.4 and §F.1, {en, sw, id, et, ht, ko, el, ru, bg, uk, kk} with {10%, 7%, 10%, 9%, 6%, 9%, 12%, 10%, 9%, 10%, 8%} ratio.

For parallel training data for the MT task, we use bitext from the NLLB dataset (Schwenk et al., 2021b; Heffernan et al., 2022; NLLB Team et al., 2022)[8] This includes web-scraped data, which has the potential to include nosise such as text being automatically identified as the wrong language, mis-aligned or mis-translated segments, and low-quality machine translated segments (Khayrallah & Koehn, 2018; Caswell et al., 2020; Dodge et al., 2021; Kreutzer et al., 2022; Thompson et al., 2024a). We use LASER3 (Artetxe & Schwenk, 2019) to select higher quality segments for fine-tuning. LASER has been used extensively to both locate parallel segments on the web (Schwenk et al., 2021a;b) as well as for filtering noisy sentence and document pairs (Chaudhary et al., 2019; Koehn et al., 2020; Thompson & Koehn, 2020; Sloto et al., 2023).

In adapter training for VocADT, we use a (peak) learning rate of 2e-6 with a cosine scheduler, a maximum sequence length of 512 tokens, a warm-up ratio of 0.01, and a weight decay of 0.01. In full-weight fine-tuning phase, we mostly follow the training setting from ALMA.

**Details of Baseline.**   For ZeTT, we use multilingual hypernetwork for Mistral-7B.[9] We use the code of OFA[10] and FOCUS[11] to create new embeddings for Mistral-7B.

**Machine Translation Metrics.**   We assess translation quality using xCOMET-XL (Guerreiro et al., 2023), as recent WMT metric shared tasks (Freitag et al., 2023; 2024) have found neural metrics like Yisi (Lo, 2019; Lo & Larkin, 2020), Bert-score (Zhang et al., 2019), Prism (Thompson & Post, 2020a;b), Comet (Rei et al., 2020), BLEURT (Sellam et al., 2020) correlate much better with human judgements, than surface-form metrics like BLEU (Papineni et al., 2002) or chrF (Popović, 2015; 2017) which consider only surface form. Trained metrics like the comet and BLEURT, which train on prior human annotations of translation quality, achieve the highest correlation with human judgments. While these correlations are less strong out of domain (relative to the domains used in WMT, e.g. FLORES) the trained metrics still outperform surface level ones (Zouhar et al., 2024).

We also caveat that xCOMET-XL does not consider context when judging translation quality, and context has been shown to be an important aspect of translation quality evaluation (for a thorough overview see (Castilho & Knowles, 2024), especially for LLMs Karpinska & Iyyer (2023). While there have been several efforts to incorporate context in MT evaluation (e.g. Vernikos et al. (2022); Deutsch et al. (2023); Raunak et al. (2024), there is no consensus in the community as to which method, so we stick to the established xCOMET-XL at the sentence level. Finally, metric differences, especially small ones, may not correspond to statistically significant differences (Koehn, 2004; Deutsch et al., 2021; Lo et al., 2023; Thompson et al., 2024b).

---

[8]https://huggingface.co/datasets/allenai/nllb
[9]https://github.com/bminixhofer/zett
[10]https://github.com/cisnlp/ofa
[11]https://github.com/konstantinjdobler/focus

# D  COMPUTATIONAL COST VOCADT

We report the computational cost of our approach. We use a batch size of 128 (four A100 GPUs * 8 batch size * 4 gradient accumulation) and a sequence length of 512. The FLOPs per token for VocADT is 17.7GFLOPs/token, resulting in 1160T "FLOPs per batch" (128 * 512 * 17.7G). Our training requires 38k update steps (2.49B, roughly 0.5B per langs). Therefore, the computational cost of a VocADT model (1160T "FLOPs per batch" x 38k step). We use `profile()` method of `accelerator` Python library and our estimation of the FLOPs per token for Mistral-7B is 14.2 GFLOPs/token.

# E  LANGUAGE-WISE RESULTS OF VOCABULARY ADAPTATIONS

Table 3: `xx-en` MT results with xCOMET-XL score and the increase rate from the original Mistral after the vocabulary adaptation—only replacing embeddings while fixing the rest.

| MT (`xx-en`) | Mistral | VocADT-multi (Ours) | | ZETT-mono | | OFA-multi | | FOCUS-mono | |
|---|---|---|---|---|---|---|---|---|---|
| `sw-en` | 0.485 | 0.801 | 65.32% | 0.734 | 51.50% | 0.215 | -55.75% | 0.216 | -55.53% |
| `id-en` | 0.946 | 0.942 | -0.44% | 0.880 | -7.01% | 0.246 | -73.97% | 0.186 | -80.36% |
| `et-en` | 0.722 | 0.899 | 24.46% | 0.769 | 6.40% | 0.196 | -72.93% | 0.248 | -65.70% |
| `ht-en` | 0.554 | 0.669 | 20.72% | 0.484 | -12.55% | 0.249 | -54.96% | 0.212 | -61.72% |
| `ko-en` | 0.882 | 0.755 | -14.39% | 0.662 | -24.87% | 0.189 | -78.56% | 0.318 | -63.99% |
| `el-en` | 0.438 | 0.760 | 73.59% | 0.664 | 51.69% | 0.182 | -58.54% | 0.250 | -42.87% |
| `ru-en` | 0.959 | 0.927 | -3.33% | 0.882 | -8.06% | 0.249 | -74.06% | 0.264 | -72.43% |
| `bg-en` | 0.952 | 0.918 | -3.56% | 0.896 | -5.93% | 0.228 | -76.00% | 0.271 | -71.55% |
| Avg (8 pairs) | 0.742 | 0.834 | 12.35% | 0.746 | 0.56% | 0.219 | -70.47% | 0.246 | -66.92% |
| `uk-en` | 0.944 | 0.915 | -3.07% | – | – | 0.201 | -78.70% | 0.288 | -69.49% |
| `kk-en` | 0.411 | 0.763 | 85.82% | – | – | 0.190 | -53.72% | 0.308 | -24.98% |
| Avg (10 pairs) | 0.729 | 0.835 | 14.49% | – | – | 0.215 | -70.59% | 0.256 | -64.89% |
| Total # of Models | 1 | 3 | | 8 | | 3 | | 10 | |

Table 4: `en-xx` MT results with xCOMET-XL score and the increase rate from the original Mistral after the vocabulary adaptation—only replacing embeddings while fixing the rest.

| MT (`en-xx`) | Mistral | VocADT-multi (Ours) | | ZETT-mono | | OFA-multi | | FOCUS-mono | |
|---|---|---|---|---|---|---|---|---|---|
| `en-sw` | 0.238 | 0.562 | 135.88% | 0.673 | 182.29% | 0.342 | 43.54% | 0.209 | -12.23% |
| `en-id` | 0.778 | 0.837 | 7.65% | 0.836 | 7.47% | 0.436 | -43.99% | 0.224 | -71.26% |
| `en-et` | 0.309 | 0.643 | 108.37% | 0.652 | 111.23% | 0.405 | 31.12% | 0.247 | -19.86% |
| `en-ht` | 0.308 | 0.329 | 7.03% | 0.339 | 10.11% | 0.227 | -26.38% | 0.235 | -23.61% |
| `en-ko` | 0.703 | 0.598 | -14.99% | 0.631 | -10.24% | 0.309 | -56.06% | 0.258 | -63.33% |
| `en-el` | 0.384 | 0.413 | 7.56% | 0.524 | 36.71% | 0.232 | -39.58% | 0.215 | -43.90% |
| `en-ru` | 0.900 | 0.854 | -5.17% | 0.719 | -20.10% | 0.388 | -56.87% | 0.371 | -58.80% |
| `en-bg` | 0.899 | 0.859 | -4.43% | 0.755 | -16.02% | 0.332 | -63.03% | 0.289 | -67.80% |
| Avg (8 pairs) | 0.565 | 0.637 | 12.77% | 0.641 | 13.53% | 0.334 | -40.89% | 0.256 | -54.66% |
| `en-uk` | 0.865 | 0.851 | -1.59% | – | – | 0.318 | -63.23% | 0.310 | -64.14% |
| `en-kk` | 0.222 | 0.522 | 135.11% | – | – | 0.294 | 32.25% | 0.223 | 0.65% |
| Avg (10 pairs) | 0.560 | 0.647 | 15.40% | – | – | 0.328 | -41.44% | 0.258 | -53.93% |
| Total # of Models | 1 | 3 | | 8 | | 3 | | 10 | |

Table 5: XNLI results with Accuracy score and the increase rate from the original Mistral after the vocabulary adaptation—only replacing embeddings while fixing the rest.

| XNLI | Mistral | VocADT-multi (Ours) | | ZETT-mono | | OFA-multi | | FOCUS-mono | |
|---|---|---|---|---|---|---|---|---|---|
| en | 0.550 | 0.553 | 0.47% | 0.554 | 0.73% | 0.547 | -0.47% | 0.537 | -2.30% |
| sw | 0.353 | 0.398 | 12.63% | 0.453 | 28.33% | 0.345 | -2.16% | 0.325 | -7.96% |
| el | 0.419 | 0.387 | -7.60% | 0.380 | -9.31% | 0.330 | -21.21% | 0.337 | -19.58% |
| ru | 0.488 | 0.490 | 0.48% | 0.474 | -2.87% | 0.347 | -28.98% | 0.331 | -32.27% |
| bg | 0.425 | 0.457 | 7.63% | 0.450 | 5.88% | 0.344 | -19.02% | 0.371 | -12.59% |
| Avg (5 langs) | 0.447 | 0.457 | 2.24% | 0.462 | 3.40% | 0.383 | -14.38% | 0.380 | -14.93% |
| Total # of Models | 1 | 2 | | 5 | | 2 | | 5 | |

Table 6: XCOPA results with Accuracy score and the increase rate from the original Mistral after the vocabulary adaptation—only replacing embeddings while fixing the rest.

| XCOPA | Mistral | VocADT-multi (Ours) | | ZETT-mono | | OFA-multi | | FOCUS-mono | |
|---|---|---|---|---|---|---|---|---|---|
| sw | 0.510 | 0.574 | 12.55% | 0.576 | 12.94% | 0.564 | 10.59% | 0.544 | 6.67% |
| id | 0.584 | 0.608 | 4.11% | 0.598 | 2.40% | 0.508 | -13.01% | 0.512 | -12.33% |
| et | 0.470 | 0.538 | 14.47% | 0.536 | 14.04% | 0.516 | 9.79% | 0.520 | 10.64% |
| ht | 0.514 | 0.548 | 6.61% | 0.572 | 11.28% | 0.526 | 2.33% | 0.534 | 3.89% |
| Avg (4 langs) | 0.520 | 0.567 | 9.14% | 0.571 | 9.82% | 0.529 | 1.73% | 0.528 | 1.54% |
| Total # of Models | 1 | 1 | | 4 | | 1 | | 5 | |

Table 7: Belebele results with Accuracy score and the increase rate from the original Mistral after the vocabulary adaptation—only replacing embeddings while fixing the rest.

| Belebele | Mistral | VocADT-multi (Ours) | | ZETT-mono | | OFA-multi | | FOCUS-mono | |
|---|---|---|---|---|---|---|---|---|---|
| en | 0.843 | 0.833 | -1.18% | 0.780 | -7.51% | 0.546 | -35.31% | 0.367 | -56.52% |
| sw | 0.391 | 0.440 | 12.50% | 0.476 | 21.61% | 0.248 | -36.65% | 0.252 | -35.51% |
| id | 0.647 | 0.638 | -1.38% | 0.610 | -5.67% | 0.289 | -55.33% | 0.230 | -64.43% |
| et | 0.439 | 0.538 | 22.53% | 0.380 | -13.42% | 0.250 | -43.04% | 0.213 | -51.39% |
| ht | 0.397 | 0.507 | 27.72% | 0.507 | 27.73% | 0.248 | -37.54% | 0.240 | -39.50% |
| ko | 0.666 | 0.616 | -7.52% | 0.466 | -30.05% | 0.278 | -58.27% | 0.274 | -58.77% |
| el | 0.442 | 0.566 | 27.90% | 0.468 | 5.79% | 0.287 | -35.17% | 0.284 | -35.68% |
| ru | 0.727 | 0.696 | -4.29% | 0.459 | -36.85% | 0.248 | -65.90% | 0.239 | -67.13% |
| bg | 0.674 | 0.698 | 3.47% | 0.446 | -33.93% | 0.276 | -59.14% | 0.233 | -65.40% |
| Avg (9 langs) | 0.581 | 0.614 | 5.83% | 0.510 | -12.16% | 0.296 | -48.95% | 0.259 | -55.35% |
| uk | 0.728 | 0.693 | -4.76% | _ | _ | 0.254 | -65.05% | 0.231 | -68.25% |
| kk | 0.364 | 0.442 | 21.36% | _ | _ | 0.256 | -29.87% | 0.220 | -39.63% |
| Avg (11 langs) | 0.574 | 0.606 | 5.50% | _ | _ | 0.289 | -49.70% | 0.253 | -55.93% |
| Total # of Models | 1 | 3 | | 9 | | 3 | | 11 | |

Table 8: Multilingual MMLU results with Accuracy score and the increase rate from the original Mistral after the vocabulary adaptation—only replacing embeddings while fixing the rest.

| MMMLU | Mistral | VocADT-multi (Ours) | | ZETT-mono | | OFA-multi | | FOCUS-mono | |
|---|---|---|---|---|---|---|---|---|---|
| en | 0.607 | 0.577 | -4.88% | 0.537 | -11.50% | 0.464 | -23.48% | 0.288 | -52.61% |
| id | 0.468 | 0.410 | -12.49% | 0.316 | -32.53% | 0.256 | -45.34% | 0.269 | -42.60% |
| ru | 0.500 | 0.468 | -6.39% | 0.348 | -30.34% | 0.259 | -48.25% | 0.272 | -45.55% |
| Avg (3 langs) | 0.525 | 0.485 | -7.62% | 0.400 | -23.73% | 0.326 | -37.84% | 0.276 | -47.39% |
| uk | 0.489 | 0.462 | -5.57% | _ | _ | 0.269 | -45.06% | 0.253 | -48.19% |
| Avg (4 langs) | 0.516 | 0.479 | -7.14% | _ | _ | 0.312 | -39.55% | 0.270 | -47.58% |
| Total # of Models | 1 | 3 | | 3 | | 3 | | 4 | |

Table 9: Tables of various vocabulary adaptation methods. The works in **bold** linearly combine original embeddings to generate new embeddings.

| Vocabulary Adaptation | Grouping | # Langs | External Resources | Base Model | Generative Task |
|---|---|---|---|---|---|
| **VocADT** (Ours) | multilingual (*Latin*, *Mixed*, *Cyrillic* group) | 11 | x | Mistral | MT |
| ZeTT (Minixhofer et al., 2024) | lang-specific | 26 | x | Mistral, XLM-R | x |
| **RAMEN** (Tran, 2020) | lang-specific | 6 | FastAlign, fastText | BERT, RoBERTa | x |
| **FVT** (Gee et al., 2022) | English, domain-specific | 1 (en) | x | BERT | x |
| **VIPI** (Mosin et al., 2023) | English, domain-specific | 1 (en) | x | BERT | x |
| **OFA** (Liu et al., 2024) | multilingual (all in 401k) | min 369 | ColexNet+ | XLM-R, RoBERTa | x |
| **FOCUS** (Dobler & de Melo, 2023) | lang-specific | 10 | fastText | XLM-R | x |
| MAD-X (Pfeiffer et al., 2020) | lang-specific | 16 | x | XLM-R | x |
| **WECHSEL** (Minixhofer et al., 2022) | lang-specific | 8 | fastText, bilingual dictionaries | RoBERTa, GPT-2 | x |
| **CW2V** (Mundra et al., 2024) | multilingual (all 4) | 4 | bilingual dictionaries | LLaMA2, RoBERTa | MT, summarization |
| **CLP** (Ostendorff & Rehm, 2023) | lang-specific | 1 (de) | GPT2-base w WECHSEL | GPT2, BLOOM | x |
| **CLP+** (Yamaguchi et al., 2024) | lang-specific | 4 | GPT2-base w WECHSEL | BLOOM-1/7B, TigerBot- 7B, Mistral-7B | summarization |
| Downey et al. (2023) | lang-specific & multilingual (Mixed, Uralic family) | 10 | x | XLM-R | x |

# F   ADDITIONAL EXPERIMENT FOR SCALABILITY AND GENERALIZABILITY OF VOCADT

## F.1   COMBINING LANGUAGES OF *Latin*, *Mixed*, AND *Cyrillic* INTO *All* GROUP

In Section 6.3, we observed that while grouping languages for the new vocabulary with a consistent script improves performance, script-based grouping strategies had little overall impact. This suggests that we can enhance the method's scalability for greater practicality with minimal performance tradeoffs. In this section, we explore a multilingual group with shared vocabulary at larger scales to provide better insights into scalability for multilingual setups.

We combine languages from the *Multi—Latin*, *Mixed*, and *Cyrillic* —groups into one unified set into *All*. This set comprises 11 languages—English and 10 non-English languages (Swahili, Indonesian, Estonian, Haitian, Korean, Greek, Russian, Bulgarian, Ukrainian, and Kazakh) as listed in Table 1. Following our experimental setup of 0.5B tokens per language, we train on a combined corpus of 5.5B monolingual tokens, covering all 11 languages.[12] We set $\alpha = 0$.

Tables 10 and 11 show that the *All* group follows trends similar to the initial *Latin*, *Mixed*, and *Cyrillic* setups. Figure 6 further illustrates that while the token count for the *All* group is slightly higher than that of the *Multi* group setup, it remains significantly lower than that of the original Mistral model.

---

[12]Available in `https://huggingface.co/h-j-han/Mistral-7B-VocADT-50k-All`

## F.2 GENERALIZATION TO LLAMA

We primarily conducted our experiments using the Mistral model. To validate the generalizability of our VocADT findings to other language models, we also test our approach on an additional candidate LM, LLaMA (Touvron et al., 2023).

We conducted an additional adaptation experiment using LLaMA2-7B, following the same experimental setup described in the main section. Figure 6 shows that the severity of fragmentation in LLaMA is similar to that in Mistral, with Greek being the most severely fragmented language followed by Korean. Tables 10 and 11 confirm that the performance trends observed with LLaMA are consistent with those seen in Mistral. Overall, Latin group languages benefit largely from vocabulary adaptation, while non-Latin languages in the Mixed group show minus or modest gains, except for Greek, which benefits due to its severe fragmentation. These findings validate that our method generalizes effectively to another language model.

Table 10: `xx-en` and `en-xx` MT results with xCOMET-XL score and the increase rate from the original Mistral after the vocabulary adaptation—only replacing embeddings while fixing the rest. The tables compare the *All* 11-language group versus the *Multi* groups—*Latin*, *Mixed*, and *Cyrillic* (each comprising 5 languages). We also compare the experiments using Mistral versus LLaMA as the base model.

| MT `xx-en` | Mistral | | | | | Llama | | |
|---|---|---|---|---|---|---|---|---|
| | Orig | VocADT-multi | | VocADT-**all** | | Orig | VocADT-multi | |
| Total # of Models | 1 | 3 | | **1** | | 1 | 3 | |
| `sw-en` | 0.485 | 0.801 | 65.32% | 0.775 | 59.89% | 0.359 | 0.698 | 94.43% |
| `id-en` | 0.946 | 0.942 | -0.44% | 0.919 | -2.89% | 0.954 | 0.933 | -2.20% |
| `et-en` | 0.722 | 0.899 | 24.46% | 0.851 | 17.79% | 0.496 | 0.858 | 72.98% |
| `ht-en` | 0.554 | 0.669 | 20.72% | 0.63 | 13.74% | 0.392 | 0.645 | 64.54% |
| `ko-en` | 0.882 | 0.755 | -14.39% | 0.834 | -5.41% | 0.872 | 0.776 | -11.01% |
| `el-en` | 0.438 | 0.760 | 73.59% | 0.856 | 95.44% | 0.439 | 0.777 | 76.99% |
| `ru-en` | 0.959 | 0.927 | -3.33% | 0.929 | -3.14% | 0.951 | 0.93 | -2.21% |
| `bg-en` | 0.952 | 0.918 | -3.56% | 0.92 | -3.38% | 0.941 | 0.916 | -2.66% |
| Avg (8 pairs) | 0.742 | 0.834 | 12.35% | 0.839 | 13.03% | 0.675 | 0.817 | 21.04% |
| `uk-en` | 0.944 | 0.915 | -3.07% | 0.909 | -3.74% | 0.947 | 0.897 | -5.28% |
| `kk-en` | 0.411 | 0.763 | 85.82% | 0.751 | 82.92% | 0.286 | 0.611 | 113.64% |
| Avg (10 pairs) | 0.729 | 0.835 | 14.49% | 0.837 | 14.76% | 0.664 | 0.804 | 21.08% |

| MT `en-xx` | Mistral | | | | | Llama | | |
|---|---|---|---|---|---|---|---|---|
| | Orig | VocADT-multi | | VocADT-**all** | | Orig | VocADT-multi | |
| `en-sw` | 0.238 | 0.562 | 135.88% | 0.466 | 95.48% | 0.291 | 0.367 | 26.12% |
| `en-id` | 0.778 | 0.837 | 7.65% | 0.763 | -1.89% | 0.868 | 0.872 | 0.46% |
| `en-et` | 0.309 | 0.643 | 108.37% | 0.587 | 90.12% | 0.279 | 0.581 | 108.24% |
| `en-ht` | 0.308 | 0.329 | 7.03% | 0.312 | 1.40% | 0.286 | 0.315 | 10.14% |
| `en-ko` | 0.703 | 0.598 | -14.99% | 0.631 | -10.23% | 0.669 | 0.566 | -15.40% |
| `en-el` | 0.384 | 0.413 | 7.56% | 0.635 | 65.56% | 0.297 | 0.511 | 72.05% |
| `en-ru` | 0.900 | 0.854 | -5.17% | 0.855 | -5.02% | 0.877 | 0.824 | -6.04% |
| `en-bg` | 0.899 | 0.859 | -4.43% | 0.854 | -4.96% | 0.826 | 0.825 | -0.12% |
| Avg (8 pairs) | 0.565 | 0.637 | 12.77% | 0.638 | 12.98% | 0.549 | 0.608 | 10.75% |
| `en-uk` | 0.865 | 0.851 | -1.59% | 0.83 | -4.05% | 0.83 | 0.814 | -1.93% |
| `en-kk` | 0.222 | 0.522 | 135.11% | 0.555 | 150.05% | 0.188 | 0.354 | 88.30% |
| Avg (10 pairs) | 0.560 | 0.647 | 15.40% | 0.649 | 15.79% | 0.541 | 0.603 | 11.46% |

Table 11: XNLI, XCOPA, Belebele, and MMMLU results with Accuracy score and the increase rate from the original Mistral after the vocabulary adaptation—only replacing embeddings while fixing the rest. The tables compare the *All* 11-language group versus the *Multi* groups—*Latin*, *Mixed*, and *Cyrillic* (each comprising 5 languages). We also compare the experiments using Mistral versus LLaMA as the base model.

| XNLI | | Mistral | | | | **Llama** | |
| | Orig | VocADT-multi | | VocADT-**all** | | Orig | VocADT-multi | |
| --- | --- | --- | --- | --- | --- | --- | --- | --- |
| Total # of Models | 1 | 3 | | **1** | | 1 | 3 | |
| en | 0.550 | 0.553 | 0.47% | 0.53 | -3.64% | 0.554 | 0.568 | 2.53% |
| sw | 0.353 | 0.398 | 12.63% | 0.397 | 12.46% | 0.348 | 0.378 | 8.62% |
| el | 0.419 | 0.387 | -7.60% | 0.396 | -5.49% | 0.370 | 0.382 | 3.24% |
| ru | 0.488 | 0.490 | 0.48% | 0.494 | 1.23% | 0.425 | 0.47 | 10.59% |
| bg | 0.425 | 0.457 | 7.63% | 0.435 | 2.35% | 0.424 | 0.388 | -8.49% |
| Avg (5 langs) | 0.447 | 0.457 | 2.24% | 0.451 | 0.89% | 0.424 | 0.437 | 3.00% |

| XCOPA | | Mistral | | | | **Llama** | |
| | Orig | VocADT-multi | | VocADT-**all** | | Orig | VocADT-multi | |
| --- | --- | --- | --- | --- | --- | --- | --- | --- |
| sw | 0.510 | 0.574 | 12.55% | 0.54 | 5.88% | 0.522 | 0.546 | 4.60% |
| id | 0.584 | 0.608 | 4.11% | 0.592 | 1.37% | 0.628 | 0.604 | -3.82% |
| et | 0.470 | 0.538 | 14.47% | 0.5 | 6.38% | 0.488 | 0.538 | 10.25% |
| ht | 0.514 | 0.548 | 6.61% | 0.538 | 4.67% | 0.506 | 0.526 | 3.95% |
| Avg (4 langs) | 0.520 | 0.567 | 9.14% | 0.542 | 4.33% | 0.536 | 0.5535 | 3.26% |

| Belebele | | Mistral | | | | **Llama** | |
| | Orig | VocADT-multi | | VocADT-**all** | | Orig | VocADT-multi | |
| --- | --- | --- | --- | --- | --- | --- | --- | --- |
| en | 0.843 | 0.833 | -1.18% | 0.824 | -2.29% | 0.482 | 0.456 | -5.39% |
| sw | 0.391 | 0.440 | 12.50% | 0.454 | 16.08% | 0.262 | 0.289 | 10.31% |
| id | 0.647 | 0.638 | -1.38% | 0.636 | -1.65% | 0.380 | 0.346 | -8.95% |
| et | 0.439 | 0.538 | 22.53% | 0.54 | 23.03% | 0.312 | 0.319 | 2.24% |
| ht | 0.397 | 0.507 | 27.72% | 0.522 | 31.59% | 0.287 | 0.322 | 12.20% |
| ko | 0.666 | 0.616 | -7.52% | 0.644 | -3.25% | 0.336 | 0.354 | 5.36% |
| el | 0.442 | 0.566 | 27.90% | 0.631 | 42.70% | 0.301 | 0.357 | 18.60% |
| ru | 0.727 | 0.696 | -4.29% | 0.71 | -2.30% | 0.428 | 0.378 | -11.68% |
| bg | 0.674 | 0.698 | 3.47% | 0.694 | 2.91% | 0.398 | 0.392 | -1.51% |
| Avg (9 langs) | 0.581 | 0.614 | 5.83% | 0.629 | 8.33% | 0.354 | 0.357 | 0.86% |
| uk | 0.728 | 0.693 | -4.76% | 0.682 | -6.32% | 0.398 | 0.352 | -11.50% |
| kk | 0.364 | 0.442 | 21.36% | 0.427 | 17.18% | 0.261 | 0.277 | 6.00% |
| Avg(11 langs) | 0.574 | 0.606 | 5.50% | 0.615 | 7.08% | 0.349 | 0.349 | 0.05% |

| MMMLU | | Mistral | | | | **Llama** | |
| | Orig | VocADT-multi | | VocADT-**all** | | Orig | VocADT-multi | |
| --- | --- | --- | --- | --- | --- | --- | --- | --- |
| en | 0.607 | 0.577 | -4.88% | 0.561 | -7.53% | 0.452 | 0.415 | -8.19% |
| id | 0.468 | 0.410 | -12.49% | 0.444 | -5.17% | 0.367 | 0.296 | -19.35% |
| ru | 0.500 | 0.468 | -6.39% | 0.468 | -6.33% | 0.355 | 0.34 | -4.23% |
| Avg (3 langs) | 0.525 | 0.485 | -7.62% | 0.491 | -6.45% | 0.391 | 0.351 | -10.23% |
| uk | 0.489 | 0.462 | -5.57% | 0.463 | -5.34% | 0.346 | 0.328 | -5.20% |
| Avg (4 langs) | 0.516 | 0.479 | -7.14% | 0.484 | -6.18% | 0.38 | 0.345 | -9.21% |

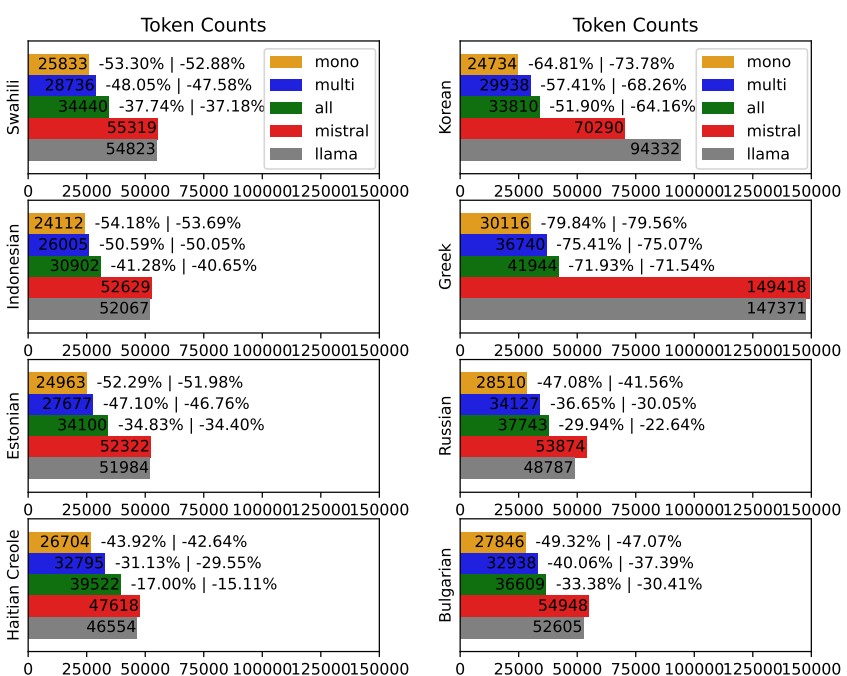

Figure 6: Token count reduction with new vocabulary. Each bar displays two percentage reduction values: the first (e.g., -53.30% in Swahili) indicates the reduction relative to the original Mistral model, while the second (e.g., -52.88%) represents the reduction relative to the original LLaMA model. We use the FLORES development set for counting the tokens by various tokenizers where the semantic contents of every language are the same. While *All* group with all 11 languages is slightly higher than that of the *Multi* group with five languages, it remains significantly lower than that of the original models. The severity of fragmentation in LLaMA is similar to that in Mistral, with Greek being the most severely fragmented language followed by Korean.

