# OpenReview forum: "Adapters for Altering LLM Vocabularies: What Languages Benefit the Most?"
_ICLR.cc/2025/Conference — ICLR 2025 Poster_

### Official Review · Reviewer_NnMw · 2024-10-20

**Soundness:** 2
**Presentation:** 2
**Contribution:** 3
**Rating:** 5
**Confidence:** 4

**Summary:**

This paper proposes a method to address the vocabulary adaptation problem in multilingual NLP, and investigates the impacts of language scripts after vocabulary adaptation. After experiments using Mistral-7b on 11 languages, they find that the method proposed achieves better or on par with the vanilla model and the baseline methods. And the Latin-script and severe fragmented languages by Mistral-7b tokenizer benefit the most after vocabulary adaptation.

**Strengths:**

The findings about the impact of language scripts are interesting, which are the Latin-script and severe fragmented languages by Mistral-7b tokenizer benefit the most after vocabulary adaptation.

**Weaknesses:**

- [addressed]**Lack of novelty**: The method proposed is an incremental work, which combines the heuristic methods like OFA in the initialization and language adapters like MAD-X. It incorporates additional parameters and computation for the adapter and 8.5B(3.5B+5B) multilingual corpora in adaptation. Moreover, the incorporated auxiliary loss is not effective in Figure 5. The improvement seems little other than the "en-el" when using this auxiliary loss ($\alpha=0.1$).

- **Missing details and additional experiments to support the findings**: only Mistral-7b is used in this work. It is better to conduct experiments on other base models like llama or gemma to improve the generalization of the findings. The detail settings for baseline methods are unknown in this paper.

- **Writing is not clear**:
  1) Line 067: ";" --> "?"
  2) Figure 1:  "Vocab Adaptor" --> "Vocab Adapter"
  3) Line 148: "lightweight": I cannot agree with this for the additional parameters and computation from the adapter training.
  4) The Figure 3 is confusing and hard to follow: a) The y-axis of 3(a) left part are increase rate(%), thus the negative results are missing like "id-en", "belebele" on Indonesian? b). The mismatched legends for left and right part, which are "VocADT-multi(ours), ZETT-mono"(left) and "mono, multi, mistral"(right). These results will be better to report on Tables.

**Questions:**

1) The number of tokens to train baseline methods is unknown. Is it same with the VocADT?
2) It can be found that the language Korean also suffers from severe fragment in Figure 3, while the improvement on Korean is little. How to interpret this phenomenon?

---

> ### Author Response · Authors · 2024-11-23
>
> We appreciate the insightful feedback from the reviewer and have provided detailed responses to each question below. Should there be any follow-up questions, we would gladly engage in further discussion.​​
>
> > Q1. “The number of tokens to train baseline methods is unknown. Is it same with the VocADT?”
>
> We are using less number of tokens compared to the baseline for adapter training.
> We compare the “adapter training” for VocADT with “hypernetwork training” in ZeTT and “external embedding training” for FOCUS.
> - ZeTT: In Minixhofer et al., 2024, they train English + Code hypernetwork for 200k gradient update steps with a batch size of 128 tokens and a sequence length of 128. For the multilingual decoder-style models, we start from the English + Code checkpoint. Therefore, 6.55B tokens in total.
> - FOCUS use fastText which is trained on Wikipedia and CommonCrawl. The fastText did not state the exact numbers in their paper for all language. However, we found that Russian only (overlaps with our  Mixed and Cyrillic group) has 102B tokens in CommonCrawl and 4% of coverage in training already exceed the numbers of token that we used for adapter training, where the minimum coverage rate they reported is 63.4%.
> - OFA use ColexNet+ as their external embeddings, which is also unclear to identify the numbers of tokens they used to train the embeddings.
>
> | Ours/Baelines | VocADT | ZeTT |
> |---|---|---|
> | Token Trained | 3.5B | 6.5B |
> | Trained Part | Adapter | Multilingual Hypernetwork |
>
>
>
> > Q2. “It can be found that the language Korean also suffers from severe fragment in Figure 3, while the improvement on Korean is little. How to interpret this phenomenon?”
>
> We believe the severity of overfragmentation in Greek far surpasses that in Korean, as Greek exhibits more than twice the severity, and that this distinction makes it hard to apply the same explanation to both Korean and Greek languages.
> In Figure 3, the token count in Greek (150k, with the bars clipped) is more than twice that of Korean (70k), while Bulgarian and Russian show counts of 55k and 54k, respectively. Therefore, we group Korean’s fragmentation with Russian and Bulgarian as exhibiting similar severity, while Greek stands out as uniquely severe.
>
> > W1. “Lack of novelty; incremental work combining the heuristic methods like OFA in the initialization and language adapters like MAD-X.”
>
> We would like to highlight that there is a distinctive difference between MAD-X and ours in terms of architecture and as well as between OFA in terms of heuristic methodology, which make it hard to say VocADT is merely an incremental work from those.
> MAD-X incorporates adapters within the Transformer layers, as illustrated in their paper Figure 1. In contrast, our adapters are positioned at the bottom and top of the original model, creating a clear architectural divergence.
> Additionally, unlike OFA or FOCUS, which rely on initialization heuristics requiring external embeddings, we deliberately avoid such dependencies. Instead, we adopt a simple heuristic to initialize the adapter which is widely used among vocabulary transfer.
>
> > W2. “Missing details and additional experiments to support the findings: only Mistral-7b is used in this work…. The detail settings for baseline methods are unknown in this paper.”
>
> Thank you for the suggestions. We will add the detailed settings for baseline methods including the number of tokens used for the adaptation in the future version as well as the result of Llama experiment.
>
>
> > W3. “Writing is not clear: The Figure 3 is confusing and hard to follow: a) The y-axis of 3(a) left part are increase rate(%), thus the negative results are missing like "id-en", "belebele" on Indonesian? b). The mismatched legends for left and right part, which are "VocADT-multi(ours), ZETT-mono"(left) and "mono, multi, mistral"(right). These results will be better to report on Tables."
>
> We limited the left side of Figure 3 to the positive range to make it easier to see which languages benefit from vocabulary transfer, but as you mentioned, it may be confusion, so we also include full tables in Appendix B.

---

> > ### Comment · Reviewer_NnMw · 2024-11-26
> > **Response to Authors**
> >
> > Thanks for your responses, which partly address my concerns. However, the experiments for other language models are still missing to support the generalization of your method. I have updated my reviews for your responses.

---

> > > ### Author Response · Authors · 2024-12-04
> > >
> > > Thank you for your responses and positive feedback. In the general response, we addressed concerns about the generalization of our method on another language model. We genuinely value the time and effort you have invested in reviewing our work.

---

### Official Review · Reviewer_qgGK · 2024-11-02

**Soundness:** 3
**Presentation:** 3
**Contribution:** 3
**Rating:** 6
**Confidence:** 4

**Summary:**

The article introduces VocADT, a method to conduct vocabulary adaptation with adapters. This method enables the training of new vocabularies while keeping all parameters of the original model fixed. Experimental results demonstrate that VocADT outperforms existing baselines. The study finds that languages with Latin scripts and those with severe fragmentation benefit the most from vocabulary adaptation. Additionally, the article validates that incorporating new languages through vocabulary adaptation improves performance in machine translation tasks.

**Strengths:**

1. The method proposed in this article allows for vocabulary adaptation by keeping the original model completely frozen and training only the adapter, which is highly efficient.

2. Experimental results demonstrate that the method presented in this paper can effectively improve the performance.

3. The article provides a detailed analysis of the MT task, offering a more accurate assessment of the model on cross-lingual tasks.

**Weaknesses:**

1. The article does not clearly explain or analyze the advantages of VocADT compared to heuristic-based methods and those relying on external embeddings or networks. This causes a disconnect between the claims about existing methods' shortcomings in the introduction and the proposed method.

2. In VocADT, the initialization of embeddings and the use of auxiliary loss are very similar to existing work, raising concerns about the novelty of the paper.

3. The design of the adapter in this paper assumes that the new embeddings are solely linear combinations of the old embeddings. This assumption could negatively impact performance. For example, languages with Latin scripts may perform better under this assumption, while languages with greater differences, such as Korean, might not fit this assumption effectively.

**Questions:**

1. Can you further explain the limitations of heuristics-based methods and methods that rely on external embeddings? For example, what does it mean when the introduction states that "heuristics-based methods often lack adaptability"?

2. Is your proposed method, VocADT, effective for low-resource languages?

---

> ### Author Response · Authors · 2024-11-23
>
> We appreciate the insightful feedback from the reviewer and have provided detailed responses to each question below. Should there be any follow-up questions, we would gladly engage in further discussion.​​
>
> > Q1. “Can you further explain the limitations of heuristics-based methods and methods that rely on external embeddings? For example, what does it mean when the introduction states that ‘heuristics-based methods often lack adaptability’?”
>
>  Instead of relying on methods that heuristically determine weights to initialize the new embeddings using a weighted average of the original embeddings, our approach learns these weights by training adapters.
>
> The base principle of most vocabulary transfer methods is to generate new embeddings based on the original ones. For example, methods like FVT and VIPI create the new embeddings by simply averaging the embeddings of subword tokens. Meanwhile, approaches such as WECHSEL, OFA, RAMEN, and FOCUS use external resources to calculate the weights for initializing new embeddings through a weighted average of original embeddings.
>
> However, embeddings generated by these heuristic-based methods are not fully integrated with the main body of the original model and typically require an additional training phase involving full-weight updates to adjust to the new vocabulary (i.e. LAPT). In contrast, our method creates new embedding vectors by learning how to combine the existing embedding vectors without requiring external resources. This approach enables the model, with only its embeddings replaced, to perform effectively compared to heuristic-based models.
>
> > Q2. “Is your proposed method, VocADT, effective for low-resource languages?”
>
> Based on the result, the VocADT method is effective for LR languages. We experimented with two low-resource languages, Swahili and Haitian Creole (Table 1). All the task performance for these two languages increased with VocADT methods, while ht-en translation decreased with ZeTT method (Figure 3).

---

> > ### Comment · Reviewer_qgGK · 2024-11-28
> >
> > Thank you for your response. However, it seems that not all of my questions were addressed. Was it a matter of time, or did they perhaps slip through? Depending on the situation, I may reconsider my score.

---

> > > ### Author Response · Authors · 2024-11-30
> > >
> > > We appreciate your feedback and reminder. We apologize for missing some parts and tried to cover all your points in the review below.
> > >
> > >
> > > > W1. “The article does not clearly explain or analyze the advantages of VocADT compared to heuristic-based methods and those relying on external embeddings or networks. This causes a disconnect between the claims about existing methods' shortcomings in the introduction and the proposed method.”
> > >
> > > Building on the response to Q1, the main advantage of VocADT lies in the better adaptability of new embeddings to the original language model body. Unlike existing methods that rely on heuristic-based approaches and external resources, VocADT learns the combination of existing embeddings directly. We empirically demonstrate that VocADT outperforms OFA and FOCUS in both the adaptation phase and the fine-tuning phase.
> > >
> > > We appreciate your thoughtful feedback and will enhance the introduction section to better articulate the limitations of existing methods and highlight the corresponding advantages of our approach.
> > >
> > >
> > > > W2. “In VocADT, the initialization of embeddings and the use of auxiliary loss are very similar to existing work, raising concerns about the novelty of the paper.”
> > >
> > >
> > > We would like to emphasize that the main novelty of our work lies in the adaptation methods and architectural design, specifically in how to learn a combination of existing embeddings using adapter modules, rather than in the initialization or the use of auxiliary loss. This approach is distinct from the methods that rely on heuristic methods with external resources to combine existing ones (as seen in OFA and FOCUS), or the methods that train a separate hypernetwork, as in ZeTT.
> > >
> > > The idea behind simple adapter initialization, such as copying overlapping word entries, is indeed a common practice across vocabulary transfer methods as a part of the adaptation process. We also adopt this straightforward initialization approach to set a good starting point for subsequent adaptation (Section 3.2). However, our primary contribution lies in the design of the adapter mechanisms and their role in driving the adaptation process.
> > >
> > > Regarding the auxiliary loss, which encourages adapter entries for overlapping words to remain close to their initial values, our contribution lies in demonstrating its varying effects across language groups. We empirically show that this loss is counterproductive for Latin script group languages but beneficial for non-Latin group languages.

---

> > > > ### Author Response · Authors · 2024-11-30
> > > >
> > > > > W3. “The design of the adapter in this paper assumes that the new embeddings are solely linear combinations of the old embeddings. This assumption could negatively impact performance. For example, languages with Latin scripts may perform better under this assumption, while languages with greater differences, such as Korean, might not fit this assumption effectively.”
> > > >
> > > >
> > > > Our assumption is grounded in the core idea underlying most vocabulary transfer methods: generating new embeddings based on the original embeddings. [Mundra et al](https://aclanthology.org/2024.conll-1.8/). even establish theoretically that initializing within the convex hull of existing embeddings—such as using a weighted average of source embeddings—is a good initialization. Typically, these methods begin by copying embedding vectors for identical tokens from the original to the new embedding space. Subsequently, multiple embedding vectors from the original embeddings are combined to create embeddings for new vocabulary items. For example, methods such as FVT and VIPI compute the new embeddings by simply averaging the embeddings of subword tokens, while approaches like WECHSEL, OFA, RAMEN, and FOCUS utilize external resources to determine the weights to initialize the new embeddings with a weighted average of the original embeddings. While the details of the approach differ across methods, the overarching idea remains consistent.
> > > >
> > > > However, as you pointed out, this approach may have limitations, particularly for languages where the new tokens are difficult to decompose into subwords using the original tokenizer, such as Korean.
> > > > For scripts like Hangul (script of Korean), which represent syllables or combinations of consonants and vowels rather than isolated phonemes, some tokens may fail to be fully decomposed by the original vocabulary and instead require byte-level decomposition. For instance, the VocADT-Mixed grouped model includes a new token “처럼” (meaning “like” in English), a highly frequent word in Korean. However, the original Mistral vocabulary lacks a token for “럼,” preventing the decomposition of “처럼” without relying on byte-level fallback, and a combination of byte-level token may not be a good representation of a character. (This issue typically does not arise with “alphabetic” scripts, such as Latin, Cyrillic, or Greek, as these scripts can be easily decomposed into individual alphabetic characters.)
> > > > Empirically, the performance decrease rate for Korean in vocabulary transfer is higher than for Russian or Bulgarian in the same Mixed group, even though the fragmentation improvement for Korean is greater. This limitation may help explain this discrepancy.
> > > >
> > > >
> > > > \
> > > > \
> > > > Please let us know if there are any remaining points that we may have missed or that is still not clear. We would be glad to provide further clarification or elaborate on specific aspects. Thank you once again for your engagement in the discussion of this paper.

---

> > > > > ### Author Response · Authors · 2024-12-04
> > > > >
> > > > > We appreciate your time and effort in reviewing our work. Thank you for your reviews.

---

### Official Review · Reviewer_CsRD · 2024-11-04

**Soundness:** 3
**Presentation:** 3
**Contribution:** 3
**Rating:** 8
**Confidence:** 4

**Summary:**

The authors propose a new vocabulary adaptation method that enables new words to be added to a pretrained LLM's vocabulary. The proposed approach uses adapters to learn embeddings for new words. The new embeddings are linear combinations of the existing embeddings. The approach is tested on 11 languages, showing that the best results are achieved for highly fragmented and Latin script based languages.

**Strengths:**

1. The methodology is clearly explained with reference to the prior work.
2. The experiment design of this paper is well-crafted and supports the claims of the paper.

**Weaknesses:**

1. The motivation behind restricting the new embedding to a linear combination of original embeddings has not been explained.
2. Since the approach requires some amount of training, the authors should report the computational cost of their approach compared to the baselines.
3. The results in Appendix B shows significant difference among performance gains of different languages. The authors should perform analysis to determine why this happens. Does the amount of new pretraining data used for vocabulary adaptation affect the results?

**Questions:**

1. Can you give an example where the case 3 mentioned in section 3.2 happen?

---

> ### Author Response · Authors · 2024-11-23
>
> We appreciate the insightful feedback from the reviewer and have provided detailed responses to each question below. Should there be any follow-up questions, we would gladly engage in further discussion.​​
>
> > Q1. “Can you give an example where the case 3 mentioned in section 3.2 happen?”
>
> The third scenario, where a token from the new vocabulary is neither an overlapping token nor subworded by the original tokenizer, does not occur when the original vocabulary supports byte fallback or when the script is “alphabetic,” such as Latin, Cyrillic, or Greek. These cases can easily be decomposed into byte-level components or individual alphabetic characters.
> However, for scripts like Hangul, which represent combinations of consonants and vowels rather than isolated phonemes, some tokens may fall into this category if byte-level decomposition is not feasible. For instance, the VocADT-Mixed grouped model includes a new token “처럼” (meaning “like” in English), a highly frequent word in Korean—the only non-alphabetic language among the eight we examine. However, the original Mistral does not have a token for “럼,” making it impossible to decompose the new word without relying on byte-level decomposition. Most modern language models now support byte-level decomposition, which makes this third case increasingly uncommon in practice.
>
> > W1. “The motivation behind restricting the new embedding to a linear combination of original embeddings has not been explained.”
>
> The fundamental principle behind most vocabulary transfer methods is to generate new embeddings based on the original embeddings. Typically, these methods begin by copying embedding vectors for identical tokens from the original to the new embedding space. Subsequently, multiple embedding vectors from the original embeddings are combined to create embeddings for new vocabulary items. For example, methods such as FVT and VIPI compute the new embeddings by simply averaging the embeddings of subword tokens, while approaches like WECHSEL, OFA, RAMEN, and FOCUS utilize external resources to determine the weights to initialize the new embeddings with a weighted average of the original embeddings. While the details of the approach differ across methods, the overarching idea remains consistent. Mundra et al. even establish theoretically that initializing within the convex hull of existing embeddings—such as using a weighted average of source embeddings—is a good initialization.
>
> Our motivation stems from the question: rather than deciding how to combine existing embedding vectors heuristically, why not learn this process to create new embedding vectors? Relying on heuristics may lack adaptability that typically requires an additional training phase of full-weight updates to fully adapt to the new vocabulary. Building upon prior works, we propose to learn linear combinations with vocabulary adapters. We will further elaborate on this motivation in a future version.

---

> > ### Author Response · Authors · 2024-11-23
> >
> > > W2. “Since the approach requires some amount of training, the authors should report the computational cost of their approach compared to the baselines.”
> >
> > We will report the computational cost of our approach with a comparison in the next version of the paper.
> > Our computational cost of a VocADT model is lower than that of the baseline when comparing an adapter training for VocADT with a multilingual hypernetwork training in ZeTT.
> >
> > In the last paragraph of Section 5 and Table 8 in ZeTT paper, they report that “FLOPs per batch” (or update) of hypernet training for Mistral-7B will be 317T (128 batch size* 128 seq len * 15.4GFLOPs/token=252T for Mistral + 65T for multilingual, not english&code, hypernetwork). They mentioned that training a multilingual hypernetwork takes 400k gradient update step (200k for English&Code + 200k for multilingual starting from English&code).
> >
> > For ours, we use a batch size of 128 (four A100 gpus * 8 batch size * 4 gradient accumulation) and a sequence length of 512. The FLOPs per token for VocADT is 17.7GFLOPs/token, resulting in 1160T “FLOPs per batch” (128 * 512 * 17.7G). However, our training requires only 50k update steps.
> >
> > Therefore, the computational cost of a ZeTT multilingual hypernetwork training (317T “FLOPs per batch” x 400k step) is larger than that of a VocADT model (1160T “FLOPs per batch” x 50k step).
> >
> > Please note that there is a difference in the FLOPs per token estimation for Mistral-7B between our calculations and those reported in the ZeTT paper. ZeTT lists 15.4 GFLOPs/token in Table 8, while our estimation is 14.2 GFLOPs/token. This discrepancy may arise from the different estimation tools used: ZeTT estimate FLOPs on the basis of XLA-compiled instructions using Jax, whereas we use accelerate profiling. However, this 1.2 GFLOPs/token difference does not affect the overall conclusion. Even if we calculate 19 GFLOPs/token for VocADT roughly adjusting that difference, we get 1245T “FLOPs per batch” x 50k step, where the computational cost remains below that of ZeTT (317T “FLOPs per batch” x 400k step).
> >
> > For other baselines using external embeddings (such as OFA or FOCUS), an exact comparison of training computation between VocADT and external embedding training is challenging. This is mainly because the computational costs for training external embeddings, such as ColexNet+ and fastText, are not transparently documented, making direct comparisons impossible.
> >
> > > W3 “The results in Appendix B shows significant difference among performance gains of different languages. The authors should perform analysis to determine why this happens. Does the amount of new pretraining data used for vocabulary adaptation affect the results?”
> >
> > Our analysis concludes that the type of script used by a language alongside the its original performance contribute to the observed outcomes.
> >
> > We do not think the amount of data used for vocabulary adaptation directly correlates with performance gains of individual languages. This is evident from the sampling ratios reported in Appendix A, which show little connection to performance improvements (e.g., the high sampling ratio for Indonesian does not translate into a high performance gain). Another explanation is that higher-performing Mistral languages tend to decline, while lower-performing ones often improve, partially explaining the results. However, the original performance does not consistently align with the rate of improvement, especially among languages with similar initial scores but different scripts. For example, in Table 4, en-id (Indonesian is a Latin-script language) initially scored 0.778 and improved by 7.65% with VocADT, whereas en-ko (Korean is a non-Latin script language) started with a slightly lower score of 0.703 but decreased by -14.99%.
> >
> > We attribute these variations to the script type of the language, which seems to affect performance changes.  This explanation can also be generalized to other vocabulary transfer methods, such as those used in ZeTT, as illustrated in Figure 3.

---

> > > ### Comment · Reviewer_CsRD · 2024-11-25
> > > **Response to Rebuttal**
> > >
> > > Thanks. The response by the authors have addressed my concerns.

---

> > > > ### Author Response · Authors · 2024-12-04
> > > >
> > > > Thank you for your positive feedback! We truly appreciate your time and effort in reviewing our work.

---

### Official Review · Reviewer_XRrF · 2024-11-09

**Soundness:** 3
**Presentation:** 4
**Contribution:** 3
**Rating:** 6
**Confidence:** 5

**Summary:**

The paper introduces VocADT, an adapter-based method for vocabulary adaptation in pre-trained language models. New embeddings for an expanded vocabulary are learned by placing adapter modules between the new vocabulary and the original language model embeddings, enabling the model to use new tokens without modifying its main weights. The weights of the adapter module are initialized by (i) copying original embeddings of overlapping tokens, (ii) averaging embeddings where partitioned tokens in the original vocabulary subset exist, and (iii) randomly initializing the rest. An auxiliary loss is introduced to keep the embeddings of overlapping tokens close to their original values, preserving established representations for these tokens in the language model. The robustness of VocADT's impact on LLM performance for downstream tasks is tested by fine-tuning all model parameters after vocabulary adaptation. Detailed results are provided, evaluating 11 languages across 3 language groups — Latin, Mixed, and Cyrillic. VocADT is compared with three baseline methods: ZeTT, FOCUS, and OFA. Results show that, across languages and tasks, VocADT outperforms the original LM (Mistral). Latin languages benefit most from vocabulary adaptation, with language grouping strategies playing a minor role. Additionally, the auxiliary loss improves performance for Mixed-group languages but not for Latin languages. Finally, following ALMA, the authors fine-tune the full model to evaluate MT performance, showing that VocADT improves performance compared to the original Mistral baseline.

**Strengths:**

1. The introduction of VocADT, an efficient, adapter-based approach to vocabulary adaptation without relying on external embeddings, is novel.
2. The comprehensive experimental evaluation covers a wide range of multilingual tasks and scripts across 11 languages. It compares the proposed method against three baselines to assess the benefits of vocabulary adaptation and the robustness of VocADT.
3. The paper is well-structured and clearly presented.
4. VocADT’s method for efficient adaptation without altering main model weights is well-motivated, especially for cross-lingual transfer and adaptation to low-resource languages.

**Weaknesses:**

1. Given that grouping Mixed-scripts and Cyrillic scripts had little impact on performance, it would have been more interesting to see if grouping all languages could have achieved identical results, demonstrating the language scalability of the method.
2. The auxiliary loss, intended to retain original embeddings for overlapping tokens, lacks analysis across alpha values; further exploration in non-Latin languages might have improved performance.
3. The paper evaluates the proposed method on Mistral alongside several baselines. However, including at least one additional candidate LM would have strengthened the results.

**Questions:**

1. Adaptation improves performance for fragmented Greek but less so for similarly fragmented Korean. Does this suggest vocabulary adaptation may not be effective for highly fragmented languages, and that other properties of Greek contribute to the improvements?
2. Was any ablation conducted comparing initialization methods (e.g., one-hot vs. multi-hot) for the adapter module and their impact on downstream task performance?
3. How effective would VocADT be for mixed-script or multilingual groups with shared vocabulary at different scales? Exploring grouping strategies in larger, mixed-language models could provide valuable insights into scalability for multilingual setups.

---

> ### Author Response · Authors · 2024-11-23
>
> We appreciate the insightful feedback from the reviewer and have provided detailed responses to each question below. Should there be any follow-up questions, we would gladly engage in further discussion.
>
> > Q1. “Adaptation improves performance for fragmented Greek but less so for similarly fragmented Korean. Does this suggest vocabulary adaptation may not be effective for highly fragmented languages, and that other properties of Greek contribute to the improvements?”
>
> We argue that vocabulary adaptation is generally effective in severely fragmented languages like Greek, and we believe the severity of overfragmentation in Korean is not similar to that in Greek, as Greek exhibits more than twice the severity, and that this distinction makes it hard to apply the same explanation to both Korean and Greek languages.
> In Figure 3, the token count in Greek (150k, with the bars clipped) is more than twice that of Korean (70k), while Bulgarian and Russian show counts of 55k and 54k, respectively. Therefore, we group Korean’s fragmentation with Russian and Bulgarian as exhibiting similar severity, while Greek stands out as uniquely severe.
>
> > Q2. “Was any ablation conducted comparing initialization methods (e.g., one-hot vs. multi-hot) for the adapter module and their impact on downstream task performance?”
>
> The only ablation experiment is with the different alpha values (0 vs 0.1). We will include additional experiments with different initialization methods including different one-hot vs multi-hot and different alpha values.
>
> > Q3. “How effective would VocADT be for mixed-script or multilingual groups with shared vocabulary at different scales? Exploring grouping strategies in larger, mixed-language models could provide valuable insights into scalability for multilingual setups.”
>
> We will further experiment to explore language scalability. Thank you for the suggestion!

---

### Meta-Review · Area_Chair_x93t · 2024-12-07

**Metareview:**

This paper presents VocADT, a method for multilingual vocabulary adaptation using adapters. Specifically, rather than altering the embedding layer of the multilingual model to create a new multilingual vocabulary, the work proposes using adapters to learn linear combinations of the existing embeddings for the new vocabulary. The comprehensive experiments show that VocADT improves performance over multiple baselines across different language types and resourcefulnesses.

Strengths:
- Well-motivated, novel approach to the issue of multilingual tokenization, and is particularly useful given that it does not modify the original model weights and introduces a limited number of new parameters in the adapter models.
- Comprehensive experimental setting with many different tasks across 11 languages, including cross-lingual tasks such as translation.
The paper is well-written, and the experiments are well-designed and clearly explained (Reviewer XRrF, CSrD).

In the rebuttal, the authors provided additional results with a second language model -- the next version of the paper should also include the additional results with Llama.

Weaknesses:
- Some of the parameters and experimental settings considered are not ablated, such as the auxiliary loss alpha and the choice to divide representations across scripts versus grouping them all together (Reviewer XRrF).
- The paper presents a limited analysis of why some languages benefit more from the approach than others (e.g., the differences seen for Greek and Korean).

The authors should also include the details raised by reviewers and clarified in the rebuttal in the next version; these include the choice to limit embedding combinations to linear ones (Reviewer CSrD, qgGK) and computational costs of the experiments (Reviewer CSrD), among others.

**Additional Comments On Reviewer Discussion:**

The authors provided detailed responses to each reviewer, including providing new results using their method with a different LM (Llama). The reviewers responded to the authors' initial response, though some did not acknowledge the general response directly.

---

### Decision · Program_Chairs · 2025-01-22

Accept (Poster)